

# Improving Arctic sea ice thickness retrieved from CryoSat-2: A comprehensive optimization of a retracking algorithm, radar penetration rate, and snow depth

5   Yi Zhou[1], Yu Zhang[1,2], Changsheng Chen[3], Lele Li[4], Danya Xu[2], Robert C. Beardsley[5], Weizeng Shao[1]

[1]College of Marine Sciences, Shanghai Ocean University, Shanghai, 201306, China;
[2]Southern Marine Science and Engineering Guangdong Laboratory (Zhuhai), Zhuhai, 519082, China
[3]School for Marine Science and Technology, University of Massachusetts-Dartmouth, New Bedford, 02744, USA
[4]University Corporation for Polar Research (UCPR), Beijing 100875, China
10   [5]Department of Physical Oceanography, Woods Hole Oceanographic Institution, Woods Hole, 02543, USA

*Correspondence to*: Yu Zhang (yuzhang@shou.edu.cn)

**Abstract.** The Arctic sea ice thicknesses retrieved from the CryoSat-2 satellite are significantly influenced by sea ice surface roughness, snow backscatter, and snow depth on the sea ice. This study is the first to improve the retrieval of sea ice thickness from CryoSat-2 data derived by the Alfred Wegener Institute (AWI CS2) through applying a comprehensive optimization of 15   an improved retracking algorithm, corrected radar penetration rate, and new snow depth. The radar freeboard data was obtained from the improved retracking algorithm of the Lognormal Altimeter Retracker Model (LARM). The radar penetration rates were corrected to 0.77 for first-year ice (FYI), 0.96 for multi-year ice (MYI), and 0.91 for all ice types. The new snow depth data was derived from the Chinese satellite Feng Yun-3B with the MicroWave Radiometer Imager (FY3B/MWRI). Three individual and one combined optimization cases were created by focusing on the retracking algorithm, radar penetration rate, 20   and snow depth, which were validated with in situ observations collected from the National Aeronautics and Space Administration Operation IceBridge (OIB), Ice Mass Balance buoys (IMB), CryoSat Validation Experiment (CryoVEX), and Alfred Wegener Institute IceBird Program (AWI IceBird). The assessment results showed that all the optimization cases had the ability to effectively improve the sea ice thickness, with similar correlation coefficients. In the validation with the four kinds of observed data, the optimization cases reduced the RMSE of AWI CS2 up to 0.23 m (25.0 %), 0.27 m (29.7 %), 0.26 25   m (25.5 %), and 0.22 m (23.9 %). The improved sea ice thickness retrieval retained the major distribution patterns generated by AWI CS2, but generally showed thinner sea ice thickness. In particular, in the MYI region, the difference in thickness became increasingly evident from fall to spring. The differences in the variation trend between the optimization cases and AWI CS2 were significant in some coastal regions and the central Arctic. The experiments revealed that the radar penetration rate calculation was more sensitive to sea ice density than to snow density. The sensitivity experiments suggested that the snow 30   depth of TOPAZ4, in addition to that of FY3B/MWRI, was also applicable in improving the retrieval of sea ice thickness. The updated scheme of sea ice densities (FYI = 925 kg m$^{-3}$ and MYI = 902 kg m$^{-3}$) can be combined with the use of comprehensive optimization to improve the retrieval of sea ice thickness. This successful optimization provided new insights into improving



the sea ice thickness retrieved from CryoSat-2 and helped to further understand and quantify the spatiotemporal variations of sea ice thickness.

# 1 Introduction

The Arctic sea ice, an important indicator of global climate change, limits the exchange of heat, water, and momentum between the ocean and atmosphere, and has significant implications for biological activity, atmospheric circulation, thermohaline circulation, marine transport, and other processes and industries (Curry et al., 1996; Rind et al., 1995; Sévellec et al., 2017; Zhang et al., 2016). Over the past decades, Arctic sea ice has declined rapidly in the extent of retreat and in the thinning of its thickness (Comiso et al., 2008; Kwok, 2018; Markus et al., 2009; Rothrock et al., 1999; Stroeve et al., 2012).

Compared with Arctic sea ice concentration, data on sea ice thickness is relatively incomplete at spatiotemporal scales. Recent advances in satellite altimetry began in 2003. Satellite-derived Arctic sea ice thickness from ERS-1 and ERS-2 is available for 1991–2003, but only covers areas south of 81.5° N (Laxon et al., 2003). The ICESat satellite data spans 2003 to 2008 with coverage of the central Arctic Ocean up to 86° N (Kwok and Cunningham, 2008). In recent years, the CryoSat-2 (Laxon et al., 2013) and ICESat-2 (Kwok et al., 2019), launched in 2010 and 2018 respectively, have covered more regions of the Arctic Ocean south of 88° N. The sea ice thickness data derived from CryoSat-2 data has been widely used to estimate the variabilities of Arctic sea ice thickness and volume (Kwok, 2018; Kwok and Cunningham, 2015; Stroeve and Notz, 2018; Tilling et al., 2018). Numerous studies have compared CryoSat-2 sea ice thickness with other observed and satellite data (Guerreiro et al., 2017; King et al., 2018; Sallila et al., 2019). CryoSat-2 sea ice thickness has also been used as assimilation data to improve the numerical results of sea ice thickness (Blockley and Peterson, 2018; Mignac et al., 2022; Schröder et al., 2019).

Various Arctic sea ice thickness products derived from the CryoSat-2 satellite have been released by the Centre for Polar Observation and Modelling (Tilling et al., 2016), Goddard Space Flight Center (Kurtz et al., 2014), NASA Jet Propulsion Laboratory (Kwok and Cunningham, 2015), Alfred Wegener Institute (Ricker et al., 2014), and Climate Change Initiative (CCI) program (Paul et al., 2018). However, due to the different retrieval algorithms used, differences in the products can cause uncertainties in the spatiotemporal variation of sea ice thickness estimates. Based on the retrieval algorithm from the hydrostatic equilibrium equation, differences in sea ice thickness are mainly due to differences in the estimation of sea ice freeboard, snow depth, and the densities of sea ice and snow.

The estimation of sea ice freeboard is dominated by radar freeboard, which is calculated through discriminating the floes and leads and retracking the radar waveform to obtain the main radar scattering interface of the floes and leads (e.g., retracking points). Currently, most products use threshold first-maximum retracker algorithms (TFMRA) to identify retracking points. TFMRA applies a fixed percentage threshold of the waveform's first maximum power return (Guerreiro et al., 2017; Laxon et al., 2013; Ricker et al., 2014; Tilling et al., 2016). However, some studies suggest that the power threshold for retracking points is affected by the sea ice surface roughness within the radar footprint scale (Kurtz et al., 2014; Landy et al., 2020; Landy et al.,





2019). Kurtz et al. (2014) first proposed a waveform fitting method, which assumed that the sea ice surface height within the radar footprint conforms to a Gaussian distribution. They concluded that the physical model-based sea ice freeboard is better than the traditional threshold methods. Moreover, Landy et al. (2020) developed a Lognormal Altimeter Retracker Model (LARM) and found that the radar freeboard derived from the LARM has minimal errors compared with the TFMRA and Gaussian physical model. All of these methods use the delayed radar signal propagation speed in the snow layer to correct sea

ice freeboard and assume that the main radar scattering interface is located at the ice-snow interface (Beaven et al., 1995).

However, many in situ observations and simulations have shown that the main radar scattering interface may be located at the snow layer due to the snow backscatter from the air-snow interface and within the snowpack (Hendricks et al., 2010; King et al., 2018; Kwok, 2014; Ricker et al., 2015; Willatt et al., 2011; Willatt, 2012). Therefore, the penetration of radar signals in the snow layer is another important correction term because it can produce systematic bias in the sea ice freeboard estimation

(King et al., 2015; Laxon et al., 2013; Ricker et al., 2014). Armitage and Ridout (2015) (AR15) defined the radar penetration factors based on the difference between CryoSat-2 measurements and in situ observations of Operation IceBridge (OIB), using this information to calculate radar penetration rates on first-year ice (FYI) and multi-year ice (MYI) by Gaussian fitting. However, this AR15 method had limitations and errors that can be summarized via three main issues, addressing which is necessary for calculating more accurate radar penetration rates. First, the definition of the radar penetration factors used an

incorrect expression in the formula for the radar propagation speed delay in the snow layer (Mallett et al., 2020). Second, the calculation of radar penetration rates is influenced by the choice of retracking algorithm. Because the radar freeboard used in AR15 is derived from an empirical threshold retracker, the radar freeboard errors were transferred to the radar penetration rates estimation. Third, the radar penetration rates were calculated based only on OIB from 2013–2014, which limited the applicable time series and regions of sea ice freeboard correction.

In addition to the sea ice freeboard, snow depth is another critical parameter in estimating sea ice thickness (Giles et al., 2008; Kern et al., 2015; Zygmuntowska et al., 2014). The snow depths used in most products are based on climatology data (Warren et al., 1999) (W99). Since the original W99 has errors, especially for FYI (Kurtz and Farrell (2011), most products modified W99 (by halving the snow depth on FYI) and then retrieved the sea ice thickness (Kwok, 2018; Kwok and Cunningham, 2015; Laxon et al., 2013; Ricker et al., 2014; Tilling et al., 2016). The CryoSat-2 product from the Alfred

Wegener Institute (AWI CS2) used new snow depth data based on combined climatology and passive microwave remote sensing (MW99/AMSR2) (Hendricks et al., 2021). With the ongoing rapid change in Arctic climate, the W99 climatology may be outdated (Webster et al., 2014) and the Arctic sea ice thickness retrieved by W99 snow depth has considerable uncertainty. More recent snow depth data based on the passive microwave, satellite altimetry, and numerical model can help to improve sea ice thickness retrieval (Blanchard-Wrigglesworth et al., 2018; Garnier et al., 2021; Kwok et al., 2020; Liston et al., 2020;

Petty et al., 2018; Rostosky et al., 2018).

The current CryoSat-2 products have the potential to enhance the accuracy of sea ice thickness determinations by modifying both sea ice freeboard and snow depth. In this study, comprehensive optimization of an improved retracking algorithm, corrected radar penetration rate, and new snow depth was used for the first time to improve the sea ice thickness retrieval of



CryoSat-2 for 2013–2018. In the retracking algorithm, we used LARM to replace TFMRA to reduce the radar freeboard errors.
For the radar penetration rate, we redefined the new radar penetration factors by correcting the radar propagation formula, using a LARM-derived radar freeboard, and expanding the spatiotemporal coverage of the measurement samples (airborne and buoy measurements during 2010–2018). For the snow depth, we developed a new dataset derived from the Feng Yun-3B satellite with the MicroWave Radiometer Imager (FY3B/MWRI). Using the three improvements above, we ran four test cases—three individual and one combined—that were compared with AWI CS2 and evaluated by OIB, Ice Mass Balance buoys (IMB), CryoSat Validation Experiment (CryoVEX), and AWI IceBird. After the assessment, we re-examined the distribution and variation patterns of Arctic sea ice thickness based on the improved data and quantify the difference from previous findings. We also discuss the potential impact of sea ice and snow densities on the calculation of radar penetration rates and retrieval of sea ice thickness, as well as the applicability of some simulated snow depth datasets to sea ice thickness retrieval.

## 2 Data and Method

### 2.1 CryoSat-2 satellite data

The CryoSat-2 is an altimetry satellite from the European Space Agency (Drinkwater et al., 2004; Wingham et al., 2006). In this study, we selected the AWI CS2 sea ice product and optimized the retracking algorithm, radar penetration rate, and snow depth to improve the retrieval of sea ice thickness. AWI CS2 has relatively complete sea ice parameters, which are necessary for our optimization cases to minimize the additional retrieval errors. Monthly mean radar freeboard, sea ice freeboard, and sea ice thickness data from AWI CS2 were used. The monthly mean radar freeboard from a LARM-derived product (Landy et al., 2020) was used to recalculate the new radar penetration rates and further evaluate the improvement of sea ice thickness retrieval. The difference between AWI CS2 and LARM-derived radar freeboard is mainly due to the different retracking algorithms, although the different classifying waveforms, geophysical corrections, and sea level tie-point interpolation also contribute to a relatively small extent (Landy et al., 2020). All the spatial resolutions were 25 km × 25 km and the period was October to April 2010–2018.

### 2.2 Snow depth data

#### 2.2.1 FY3B/MWRI

In this study, we used the daily mean snow depth data based on FY3B/MWRI to improve the retrieval of sea ice thickness from AWI CS2. The FY-3B meteorological satellite is a second-generation polar-orbiting meteorological satellite from China that was launched in November 2010. The snow depth of FY3B/MWRI was developed by Li et al. (2021) with a spatial resolution of 12.5 km × 12.5 km. The new snow depth data shows smaller biases with the in situ observations of OIB, which





had a mean difference of 2.89 cm on FYI and 1.44 cm on MYI. The period of FY3B/MWRI and the other snow depth data used in the study was from 2013 to 2018.

### 130 **2.2.2 MW99/AMSR2**

The original AWI CS2 sea ice thickness product used MW99/AMSR2, which combined the MYI snow depth from W99 and the FYI snow depth from AMSR2 with a spatial resolution of 25 km × 25 km (Hendricks et al., 2021). In this study, the MW99/AMSR2 was used in some optimization cases to analyze the improvement in sea ice thickness retrieval controlled by snow depth.

### 135 **2.2.3 Numerical models**

In addition to the four optimization cases proposed above, we also use three kinds of numerical snow depth data for sensitivity experiments to explore the impact of different snow depths on sea ice thickness retrieval. The first was from the NASA Eulerian Snow On Sea Ice Model (NESOSIM), which is a three-dimensional, two-layer (vertical), Eulerian snow budget model (Petty et al., 2018). The NESOSIM has a spatial resolution of 100 km × 100 km. The second was from the Lagrangian snow-evolution
model (SnowModel-LG), which was developed to simulate snow depth and density on a pan-Arctic scale (Liston et al., 2020). The resolution of SnowModel-LG is 25 km × 25 km. The third was from the ocean-sea ice data assimilation system of TOPAZ4, which is maintained and operated by the Copernicus Marine Environment Monitoring Service (Sakov et al., 2012). The snow depth of TOPAZ4 has a spatial resolution of 12.5 km × 12.5 km.

### **2.3 In situ observational data**

Four kinds of in situ observational data were used to calculate the radar penetration rates and validate the optimization cases, including the NASA Operation IceBridge (OIB), the Ice Mass Balance buoys (IMB), the CryoSat Validation Experiment (CryoVEX) and the Alfred Wegener Institute IceBird Program (AWI IceBird). The spatial trajectories of all in situ observations are shown in Fig. 1.

### **2.3.1 NASA Operation IceBridge (OIB)**

The OIB is an 11-year NASA airborne polar ice survey operation (Koenig et al., 2010). This mission includes snow sounding radar and an Airborne Topographic Mapper altimeter capable of providing large-scale and high-resolution snow depth and total freeboard data (sum of sea ice freeboard and snow depth). The measured snow depth and total freeboard data can be used to estimate sea ice thickness (Kurtz et al., 2013; Newman et al., 2014). Most flight tracks run from northern Greenland to Alaska during the spring (March and April). In this study, we used two versions of the OIB dataset, including the IDCSI4
(OIB L4) for the period 2011–2013 and the Quick Look (OIB Quick Look) for the period 2012–2018, to calculate radar penetration rates and validate the sea ice freeboard and thickness.

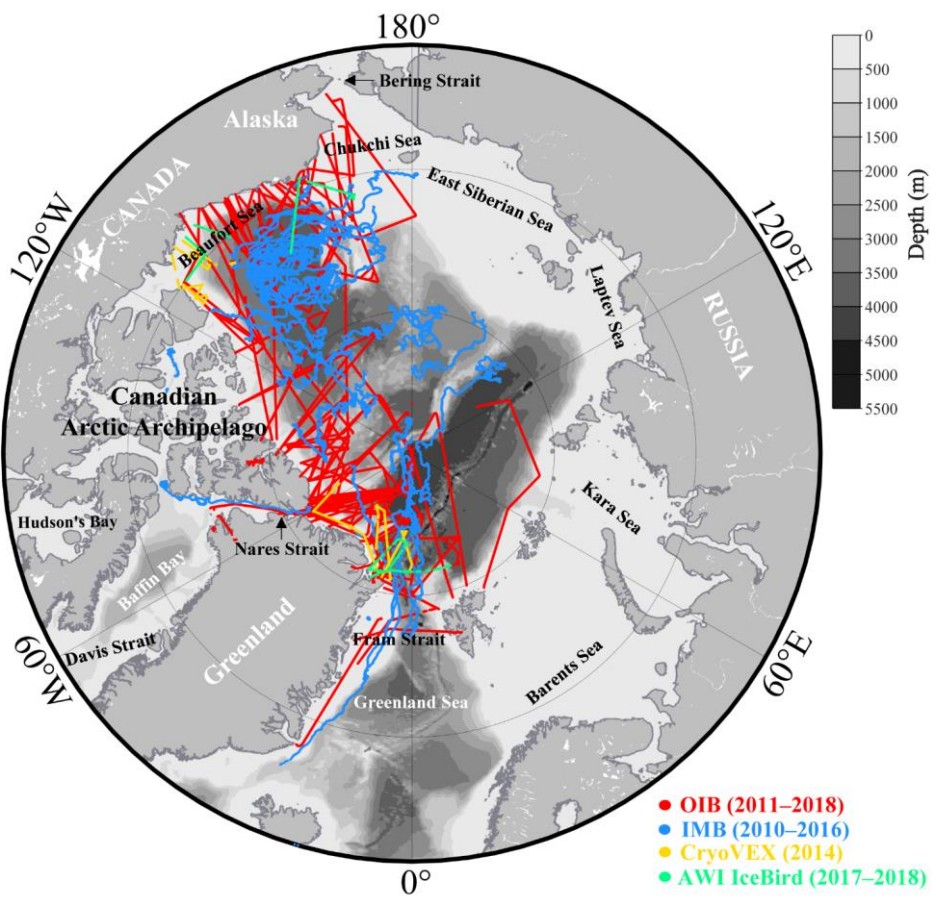

**Fig 1.** Distribution of in situ observations in the Arctic Ocean. The red, blue, yellow, and green scatter points indicate the NASA Operation IceBridge (OIB), the Ice Mass Balance buoys (IMB), the CryoSat Validation Experiment (CryoVEX), and the Alfred Wegener Institute
IceBird Program (AWI IceBird), respectively.

### 2.3.2 Ice Mass Balance buoys (IMB)

The IMB data were obtained from the Cold Regions Research and Engineering Laboratory (CRREL), which deployed sea ice mass balance buoys to monitor the mass balance and thermodynamic changes in sea ice-covered regions (Richter-Menge et al., 2005). The snow depth and sea ice thickness were collected every 4 hours. We used the buoys dataset from 2010–2018 to
calculate the radar penetration rates and validate the sea ice thickness.

### 2.3.3 CryoSat Validation Experiment (CryoVEX)

The CryoVEx is a CryoSat validation experiment performed by the European Space Agency (ESA). The aircrafts in the CryoVEx mission are equipped with airborne electromagnetic sensors and laser scanners to measure the total thickness (sum of sea ice thickness and snow depth, ice + snow). In this study, we used the total thickness data from the ESA's CryoVEX



campaign in 2014 to validate the optimization cases. The aircraft tracks are located over the Beaufort Sea and the Arctic Ocean
north of Canada (Ellesmere Island) and Greenland.

### 2.3.4 Alfred Wegener Institute IceBird Program (AWI IceBird)

The AWI IceBird program is a series of airborne measurements that primarily use an electromagnetic induction sounding
instrument, airborne laser scanner, and snow radar to measure the total thickness, total freeboard, and snow depth, respectively.
The main measurement regions are in Svalbard, Greenland, and northern Canada. Since AWI IceBird has only total thickness
data for 2018 (Jutila et al., 2021; Rohde et al., 2021), in this study we used the total freeboard and snow depth in 2017 to
calculate the radar penetration rates and used total thickness data for 2017–2018 to validate the optimization cases.

### 2.4 Data gridding

Considering all the available time coverages of the data described above, we focused on the period of 2013–2018. The satellite
altimetry and snow depth resampling data had a resolution of 25 km × 25 km and the same grids as AWI CS2. Two methods
were used for the in situ observational data. The airborne measurements (OIB, CryoVEX, and AWI IceBird) were averaged to
each AWI CS2 grid. Gridding data with less than 200 statistical points were excluded to avoid large errors (Kwok and
Cunningham, 2015; Laxon et al., 2013; Sallila et al., 2019). The drifting buoys (IMB) measure a specific ice floe, which means
that data averaged over a grid within several kilometres is not grid-representative. This can lead to noticeable error in terms of
the actual sea ice thickness. Therefore, we followed methods used in previous studies that resampled data over IMB tracks
(Bocquet et al., 2022; Kwok et al., 2007; Mu et al., 2018; Stroeve et al., 2020; Xie et al., 2018).

### 2.5 Sea ice thickness retrieval

Assuming hydrostatic equilibrium, sea ice thickness based on the CryoSat-2 satellite can be derived from the following
equation:

$h_i = (\frac{\rho_w}{\rho_w - \rho_i})h_{fi} + (\frac{\rho_s}{\rho_w - \rho_i})h_s,$           (1)

where $h_i$ is sea ice thickness, $\rho_w$, $\rho_i$ and $\rho_s$ represent sea water density, sea ice density, and snow density, respectively, $h_{fi}$ is
the sea ice freeboard, and $h_s$ is the snow depth (Fig. 2). In this study, $\rho_w$ is 1024 kg m⁻³, $\rho_i$ is 917 kg m⁻³ for FYI and 882 kg
m⁻³ for MYI. The variable $\rho_s$ is parameterized by $\rho_s = 6.50t + 274.51$ kg m⁻³ ($t$ is from 0 to 6 which represents the months
from October to April respectively) (Mallett et al., 2020). The original AWI CS2 dataset includes a low propagation speed
correction under the assumption that the radar signal completely penetrates the snow layer. The correction equation can be
expressed as:

$h_{fi}^c = h_{fr} + h_c,$           (2)





$$h_c = \left(\frac{c}{c_s} - 1\right) h_s, \tag{3}$$

where $h_{fi}^c$ is the speed-corrected sea ice freeboard, $h_{fr}$ and $h_c$ represent the radar freeboard and speed correction term,
respectively (Fig. 2), $c$ is the speed of light ($3 \times 10^8$ m s$^{-1}$), and $c_s$ is the radar propagation speed in the snow. In this study,
$c_s$ was obtained from a snow density ($\rho_s$)-dependent parameterization: $c_s = c(1 + 0.51\rho_s)^{-1.5}$ m s$^{-1}$ (Ulaby et al., 1982).

Using the sea ice freeboard retrieval based on the speed correction described above, we also considered the impact of radar
penetration (RP) on determining the main radar scattering interface. We introduced the radar penetration term to further
improve sea ice freeboard. The location of the main radar scattering interface can change the speed correction term $h_c$, which
is referred to as $h_c^p$ (Fig. 2). Based on the additional correction of radar penetration term $h_p$, the sea ice freeboard $h_{fi}^p$ is
expressed as follows:

$$h_{fi}^p = h_{fr} + h_c^p + h_p. \tag{4}$$

Therefore, the original sea ice thickness retrieval in Eq. (1) can be converted to:

$$h_i = \left(\frac{\rho_w}{\rho_w - \rho_i}\right) h_{fi}^p + \left(\frac{\rho_s}{\rho_w - \rho_i}\right) h_s. \tag{5}$$

In Eq. (4), the change of speed correction term $h_c^p$ and the radar penetration term $h_p$ are given as:

$$h_c^p = \left(\frac{c}{c_s} - 1\right) \bar{\alpha} h_s, \tag{6}$$

$$h_p = (\bar{\alpha} - 1) h_s, \tag{7}$$

where $\bar{\alpha}$ is radar penetration rate, which is the Gaussian fitting value of radar penetration factors $\alpha$ (Armitage and Ridout
(2015). Although AR15 calculated the radar penetration rates for both FYI and MYI, they are not representative due to the
incorrect expression in the formula for the radar propagation speed delay (Mallett et al., 2020). For this study, we corrected
the radar penetration factor ($\alpha$) equation from (1-C$_s$/C) to (C/ C$_s$-1) and a modified AR15 is shown below:

$$\alpha = \frac{c_s \, (h_f^{obs} - h_{fr}^{LARM})}{c \times h_s^{obs}}, \tag{8}$$

where $h_f^{obs}$ and $h_s^{obs}$ are the total freeboard and snow depth obtained from in situ observations, respectively, and $h_{fr}^{LARM}$ is the
LARM-derived radar freeboard measured by the CryoSat-2 satellite (Landy et al., 2020).

Because the in situ observation of the IMB dataset provides only sea ice thickness and snow depth data, the total freeboard
must be converted based on Archimedes' principle. The relationship of total freeboard, sea ice thickness, and snow depth is
given as follows:

$$h_f = \left(\frac{\rho_w - \rho_i}{\rho_w}\right) h_i + \left(\frac{\rho_w - \rho_s}{\rho_w}\right) h_s, \tag{9}$$

where $h_f$ is the total freeboard.





**Fig 2.** Schematic diagram of the snow-ice system based on the optimization case (left) and the original case (right). The basic variables are sea ice thickness ($h_i$), total freeboard ($h_f$), snow depth ($h_s$), sea surface height ($h_{ssh}$) and sea ice draft ($h_{draft}$). Density parameters include seawater density ($\rho_w$), sea ice density ($\rho_i$), and snow density ($\rho_s$). The height of the retracking point to the sea surface is defined as the radar freeboard ($h_{fr}$). The terms $h_c$ and $h_{fi}^c$ are speed correction and sea ice freeboard in the original case when the radar signal is considered to completely penetrate the snow layer. In the optimization case, the penetration correction term ($h_p$) is also considered, thus the speed correction term becomes $h_c^p$ and sea ice freeboard becomes $h_{fi}^p$.

## 2.6 Cases of improvement in sea ice thickness retrieval

According to Eqs. (1) and (5), the sea ice freeboard and snow depth are two key parameters in the retrieval of sea ice thickness (Zygmuntowska et al., 2014). In this study, we generated four optimization cases using the improved retracking algorithm, corrected radar penetration rate, and new snow depth to compare the improvements in the retrieval of sea ice thickness with AWI CS2. In each of the four cases, sea water, sea ice, and snow densities were consistent with those of the original AWI CS2. Case 1 (LARM + MW99/AMSR2) kept the same snow depth (MW99/AMSR2) as the original case (AWI CS2) in Eq. (5) but



used an improved retracking algorithm (LARM) to improve the radar freeboard (Table 1). Case 2 (TFMRA + MW99/AMSR2 + RP) kept the same snow depth (MW99/AMSR2) as AWI CS2 but considered radar penetration rates (RP) to improve the sea ice freeboard. Case 3 (TFMRA + FY3B/MWRI) kept the same sea ice freeboard as AWI CS2 but used the new snow depth of FY3B/MWRI to improve the parameter of snow depth. Case 4 (LARM + FY3B/MWRI + RP) was a combined case that included all the modifications in Cases 1-3 (Table 1).

**Table 1.** The sea ice thickness retrieval scheme for the 4 optimization cases and AWI CS2

| Name | Improvement | Radar penetration | Snow depth | Radar freeboard | Symbol in text |
|---|---|---|---|---|---|
| AWI CS2 | — | Ice-snow interface | MW99/AMSR2 | 50 % TFMRA | AWI CS2 (TFMRA + MW99/AMSR2) |
| Case1 | *retracking algorithm* | Ice-snow interface | MW99/AMSR2 | LARM | LARM + MW99/AMSR2 |
| Case2 | *Radar penetration* | Modified AR15 | MW99/AMSR2 | 50 % TFMRA | TFMRA + MW99/AMSR2 + RP |
| Case3 | *snow depth* | Ice-snow interface | FY3B/MWRI | 50 % TFMRA | TFMRA + FY3B/MWRI |
| Case4 | *All* | Modified AR15 | FY3B/MWRI | LARM | LARM + FY3B/MWRI + RP |

*Note that the sea water, sea ice and snow density for the 4 derived cases are consistent with AWI CS2

## 3 Assessment of improvement in sea ice thickness retrieval

### 3.1 Correction of radar penetration rates

To improve the retrieval of sea ice thickness in Cases 2 and 4, the radar penetration rates had to be recalculated based on modified AR15. The different corrected radar penetration rates on FYI, MYI, and all ice types were calculated using all in situ observations (2010–2018), OIB Quick Look (2013–2014) (which has the same period as the original AR15), OIB L4 (2011–2013), OIB Quick Look (2012–2018), AWI IceBird (2017), and IMB (2010–2016) are shown in Fig. 3. There are remarkable differences between the radar penetration rates based on different in situ observations. Compared with the radar penetration rates from the original AR15 (0.96 for FYI, 0.82 for MYI, 0.84 for all ice types), the corrected radar penetration rates on FYI are smaller, except for the calculated rate (0.97) through OIB L4 (Fig. 3c). The smallest rate (0.64) on FYI was calculated through IMB (Fig. 3f). Conversely, the corrected radar penetration rates for MYI and all ice types were large, except for the calculated rate (0.76) on MYI through AWI IceBird (Fig. 3e). Both of the largest rates on MYI and all ice types were 0.97 calculated through OIB L4. The rate on MYI calculated through IMB also showed the same large value.

The differences in radar penetration rates can be explained by the different measuring periods, spatial resolution, and performance of the instruments. For example, for the OIB, the radar penetration rates may be applicable only in the spring. The uncertainty in OIB Quick Look is different from that of OIB L4 due to the different processing methods (Kurtz et al., 2013). For the AWI IceBird, the sample points for deriving radar penetration rates are relatively few, and only in April 2017. For the IMB, although the observed data was extended to the entire wintertime, there were still disparities when compared to





airborne observations due to the different resampling methods. Some previous studies have shown that the salt-wetness of the snow layer in FYI can restrict radar penetration (Kurtz et al., 2014; Nandan et al., 2017) and reduce the radar penetration rate
on FYI relative to MYI. Landy et al. (2022) used a total penetration rate of 0.90 on all ice types to retrieve a year-round sea ice thickness from CryoSat-2. Based on the considerations above, we selected corrected radar penetration rates (0.77 for FYI, 0.96 for MYI, 0.91 for all ice types) calculated using all in situ observations. The relationship between FYI and MYI penetration rates supports the previous studies, and the calculated penetration rate for all ice types is consistent with Landy et al. (2022).

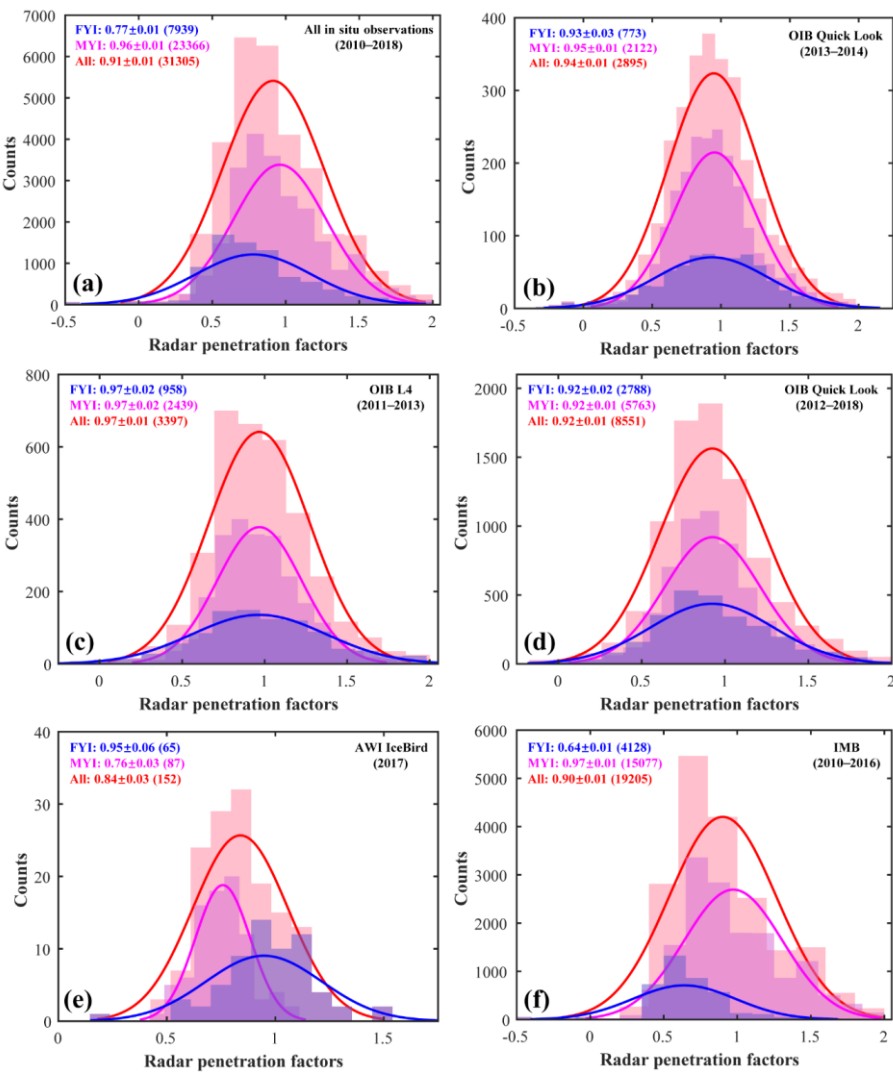

**Fig 3.** Corrected radar penetration factors histogram. The blue, purple, and red histograms indicate the radar penetration factors derived from the FYI, MYI, and all ice types, respectively. The curves represent the Gaussian fitting lines. The radar penetration rates are given in the upper left corner of the subfigure, expressed as modal value ± uncertainty (sampling number). The radar penetration rates derived from different in situ observations include (a) all in situ observations (2010–2018), (b) OIB Quick Look (2013–2014), (c) OIB L4 (2011–2013),
(d) OIB Quick Look (2012–2018), (e) AWI IceBird (2017), and (f) IMB (2010–2016).





## 3.2 Comparison of derived snow depth, sea ice freeboard, and sea ice thickness

The monthly mean snow depth, sea ice freeboard, and sea ice thickness derived from the optimization cases in Table 1 during 2013–2018 are compared in Fig. 4. The results show that the snow depth of FY3B/MWRI differed significantly from that of MW99/AMSR2. The sea ice freeboard and thickness in the different optimization cases had the same variation patterns as AWI CS2 but had considerably different values.

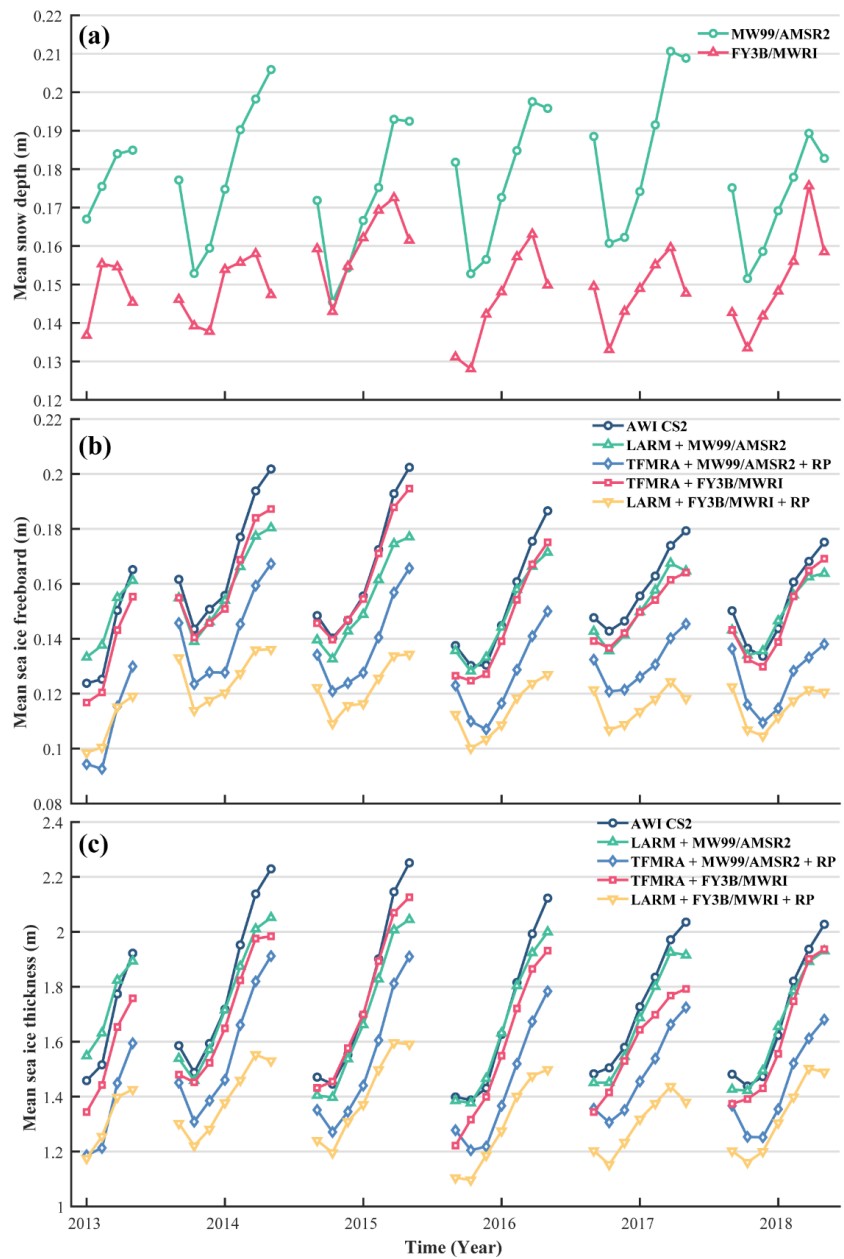

**Fig 4.** Comparisons of the monthly mean (a) snow depth, (b) sea ice freeboard, and (c) sea ice thickness in all the cases during 2013–2018.





In a comparison of snow depths (Fig. 4a), the range of FY3B/MWRI was approximately 0.13–0.17 m while that of MW99/AMSR2 was approximately 0.15–0.21 m. The mean snow depth of FY3B/MWRI was 0.03 m smaller than the snow depth of MW99/AMSR2 for the entire period. In general, the largest snow depth difference was in spring, reaching up to 0.06 m in April 2017. In addition, MW99/AMSR2 showed a noticeable decrease (about 0.02–0.03 m) from October to November each year, while the decrease of FY3B/MWRI was smaller.

The sea ice freeboard in all the optimization cases focused on a range of 0.10–0.20 m. The optimization cases generally had smaller values (0.12–0.17 m) than the original case of AWI CS2 (0.18 m) (Fig. 4b). The AWI CS2, LARM + MW99/AMSR2, and TFMRA + FY3B/MWRI had a large sea ice freeboard, and the differences between the three cases were small. The TFMRA + MW99/AMSR2 + RP and LARM + FY3B/MWRI + RP, which considered the radar penetration, had relatively small freeboard values. We also noted that the freeboard growth rate of LARM + FY3B/MWRI + RP from December to April was very low, with a growth rate of only 36–83 % of the other optimization cases and AWI CS2.

The sea ice thickness in all the cases had a range of 1.10–2.30 m and showed a variation pattern similar to that of the sea ice freeboard (Fig. 4c). The sea ice thickness differences in all the cases were larger in spring and relatively minor in winter.

**3.3 Validation of sea ice freeboard, thickness and total thickness**

Before evaluating the improvements in the retrieval of sea ice thickness, we first validated sea ice freeboard improvements by comparing them with the observations of OIB Quick Look during 2013–2018. The results show that all the optimization cases can effectively reduce the root mean square error (RMSE) of the original AWI CS2 by 0.51–3.27 cm and had a correlation coefficient (R) similar to that of OIB Quick Look (Fig. 5). The AWI CS2 generally had larger freeboard values than the OIB Quick Look (RMSE =10.35 cm and R = 0.66). The LARM + FY3B/MWRI + RP showed the most significant improvement, with an RMSE of 7.08 cm, which reduces the error by ~32 %. The annual mean sea ice freeboard improvements were also validated (Table 2). All of the optimization cases were an improvement over AWI CS2 in every year except 2016. The number of samples in 2016 was much less than in other years. The correlation coefficient in the original case of AWI CS2 was low (R = 0.24), which suggests that the satellite sea ice thickness of CryoSat-2 had greater uncertainty than OIB Quick Look. In other years, the RMSE of sea ice freeboard was reduced by 1.00–4.54 cm through the improved retracking algorithm (LARM + MW99/AMSR2), 0.13–2.55 cm through the corrected radar penetration rate (TFMRA + MW99/AMSR2 + RP), 1.00–1.71cm through the new snow depth (TFMRA + FY3B/MWRI), and 1.65–5.78 cm through the combined optimization (LARM + FY3B/MWRI + RP).

Using the improved sea ice freeboard, we validated sea ice thickness for all cases with the in situ observations of OIB Quick Look and IMB. The validation of total thickness (ice + snow) in all the cases was compared with the observational data of CryoVEX and AWI IceBird (Fig. 6). The original case of AWI CS2 had higher correlations with OIB Quick Look (R = 0.63) and CryoVEX (R = 0.74) but lower correlations with IMB (R = 0.13) and AWI IceBird (R = 0.28). The assessment showed that all the optimization cases with the improved retracking algorithm, corrected radar penetration rate, and new snow depth were successful and can effectively improve the sea ice thickness and the total thickness of AWI CS2.





In validation with OIB Quick Look, IMB, CryoVEX, and AWI IceBird, the optimization cases had similar correlation coefficients and reduced the RMSE of AWI CS2 by up to 25.0 %, 29.7 %, 25.5 %, and 23.9 %, or by 0.23, 0.27, 0.26 and 0.22 m, respectively. The LARM + FY3B/MWRI + RP and LARM + MW99/AMSR2 showed the most remarkable improvement of sea ice thickness in the validation with OIB Quick Look and IMB. The TFMRA + FY3B/MWRI had the most significant

improvement in total thickness in the validation with CryoVEX and AWI IceBird. For the optimization case of the improved retracking algorithm (LARM + MW99/AMSR2), the improvement of sea ice thickness and total thickness has the largest RMSE reduction of 0.27 and 0.25 m in the IMB and CryoVEX. Especially in the validation with IMB, the original AWI CS2 with the sea ice thickness of 3–5 m has the largest error, while the sea ice thickness of LARM + MW99/AMSR2 is decreased and the improved data is closer to IMB. In the case with the corrected radar penetration rate (TFMRA + MW99/AMSR2 +

RP), the distribution patterns of the scatter points were very similar to AWI CS2, with similar fitting slopes and similar increases in the fit intercepts. This shows that TFMRA + MW99/AMSR2 + RP decreased the sea ice thickness of AWI CS2 while maintaining the original relationship between satellite and in situ observations. The most significant improvement was in CryoVEX, which effectively reduced the RMSE by 0.15 m or ~15 %. The optimization case with new snow depth (TFMRA + FY3B/MWRI) also showed scatter point distribution patterns similar to AWI CS2. The TFMRA + FY3B/MWRI reduced

the RMSE by over 0.20 m in the validation with OIB QuickLook, CryoVEX, and AWI IceBird. Compared with the three individual optimization cases above, the combined optimization case (LARM + FY3B/MWRI + RP) generally showed similar improvements, with the largest RMSE reduction (0.23 m) with OIB Quick Look.

**Table 2.** The annual validation of sea ice freeboard with OIB Quick Look during 2013–2018

| Time (Year) | Numbers | AWI CS2 | | Case1 | | Case2 | | Case3 | | Case4 | |
|---|---|---|---|---|---|---|---|---|---|---|---|
| | | R | RMSE (cm) | R | RMSE (cm) | R | RMSE (cm) | R | RMSE (cm) | R | RMSE (cm) |
| 2013 | 1634 | 0.71 | 7.79 | 0.67 | 6.77 | 0.70 | 7.66 | 0.72 | 6.23 | 0.69 | 6.14 |
| 2014 | 2020 | 0.74 | 13.00 | 0.69 | 8.46 | 0.75 | 12.04 | 0.74 | 11.62 | 0.71 | 7.22 |
| 2015 | 1158 | 0.56 | 11.80 | 0.52 | 7.34 | 0.56 | 11.52 | 0.57 | 10.80 | 0.53 | 7.50 |
| 2016 | 391 | 0.24 | 10.00 | 0.16 | 9.85 | 0.21 | 12.55 | 0.29 | 10.51 | 0.18 | 12.95 |
| 2017 | 2304 | 0.58 | 9.63 | 0.47 | 8.50 | 0.58 | 9.04 | 0.58 | 7.92 | 0.51 | 7.21 |
| 2018 | 1375 | 0.68 | 8.33 | 0.66 | 7.33 | 0.70 | 7.02 | 0.63 | 6.85 | 0.67 | 5.20 |



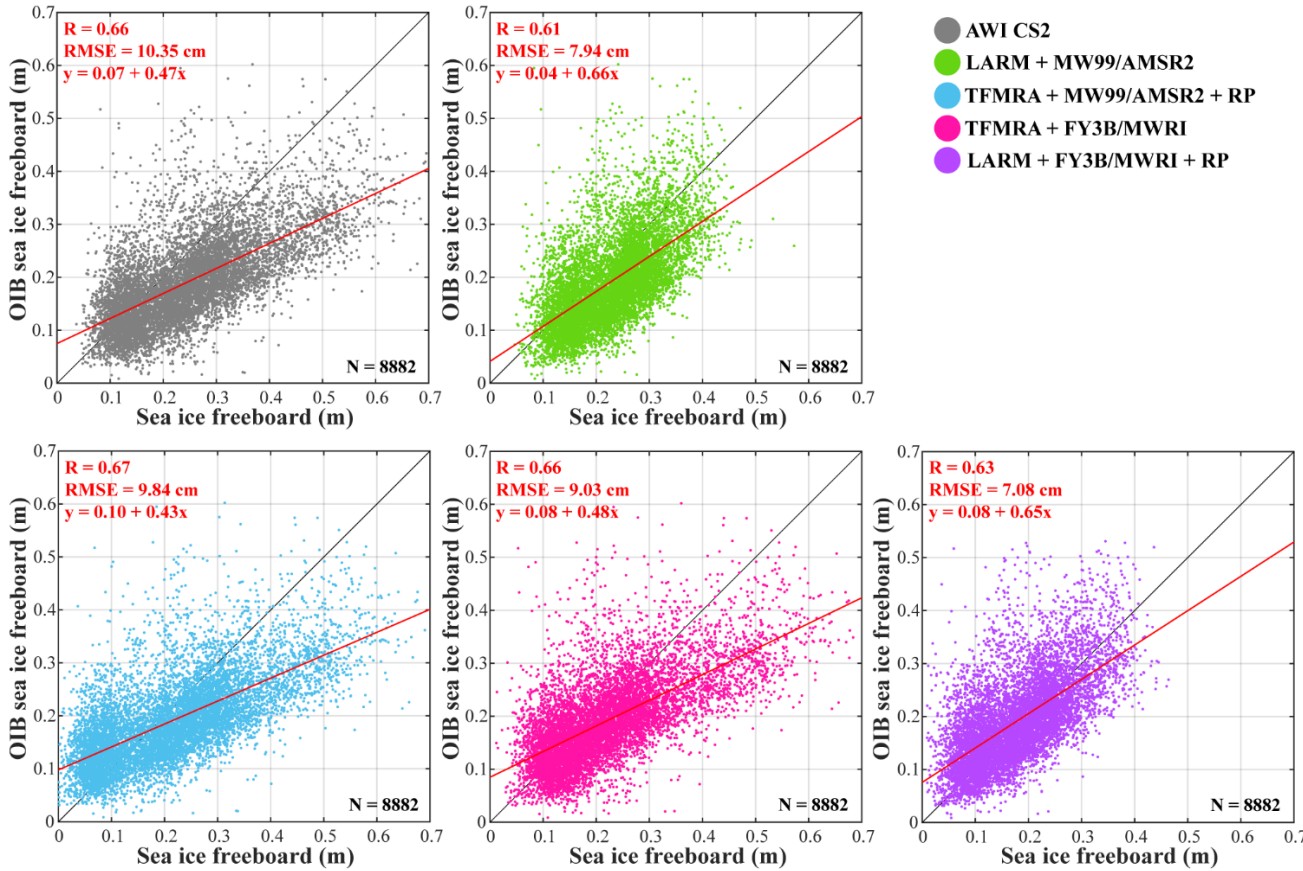

**Fig 5.** Validation of sea ice freeboard improvement with OIB Quick Look (2013–2018). The correlation coefficient (R), root mean square error (RMSE), and the number of samples (N) are shown in each subfigure. The solid black line indicates the best fitting line, and the solid red line indicates the scatter fitting line (the fitting equation is also shown in each subfigure).



**Fig 6.** Validation of sea ice thickness and total thickness (ice + snow) improvement with the in situ observational data of OIB Quick Look (2013–2018), IMB (2013–2016), CryoVEX (2014), and AWI IceBird (2017–2018). The correlation coefficient (R), root mean square error (RMSE), and the number of samples (N) are shown in each subfigure. The solid black line indicates the best fitting line, and the solid red line indicates the scatter fitting line (the fitting equation is also shown in each subfigure).





## 4 Spatio-temporal patterns of sea ice thickness based on improved data

### 4.1 Multi-year mean spatial distributions

Compared with the AWI CS2, the optimization cases had consistent patterns of sea ice thickness distribution with greater thickness north of Greenland and the Canadian Arctic Archipelago and smaller thickness in the areas of Baffin Bay, Laptev Sea, Kara Sea, and Barents Sea. However, the improved data generally showed thinner sea ice thickness than the original estimation of AWI CS2 (Fig. 7). Over the entire Arctic region (north of 65° N), the multi-year (2013–2018) mean sea ice thickness of the AWI CS2 was 1.70 m, which is larger than the results of LARM + MW99/AMSR2 (1.63 m), TFMRA + MW99/AMSR2 + RP (1.43 m), TFMRA + FY3B/MWRI (1.63 m), and LARM + FY3B/MWRI + RP (1.29 m).

The difference in sea ice thickness between the original case and optimization cases varied by region. Except for the central Arctic, some optimization cases had a larger sea ice thickness than AWI CS2 in other sub-regions (Fig. 7g). The opposite results occurred with the optimization cases with the improved retracking algorithm (LARM + MW99/AMSR2) and new snow depth (TFMRA + FY3B/MWRI). Compared with AWI CS2, the LARM + MW99/AMSR2 derived ~7–13 % greater sea ice thickness in the East Siberian Sea, Laptev Sea, Beaufort Sea, and Chukchi Sea. The TFMRA + FY3B/MWRI showed ~5–18 % greater sea ice thickness in the Kara Sea, Barents Sea, and Baffin Bay. The cases that included radar penetration (TFMRA + MW99/AMSR2 + RP, LARM + FY3B/MWRI + RP) maintained smaller thicknesses than AWI CS2 in each sub-region.

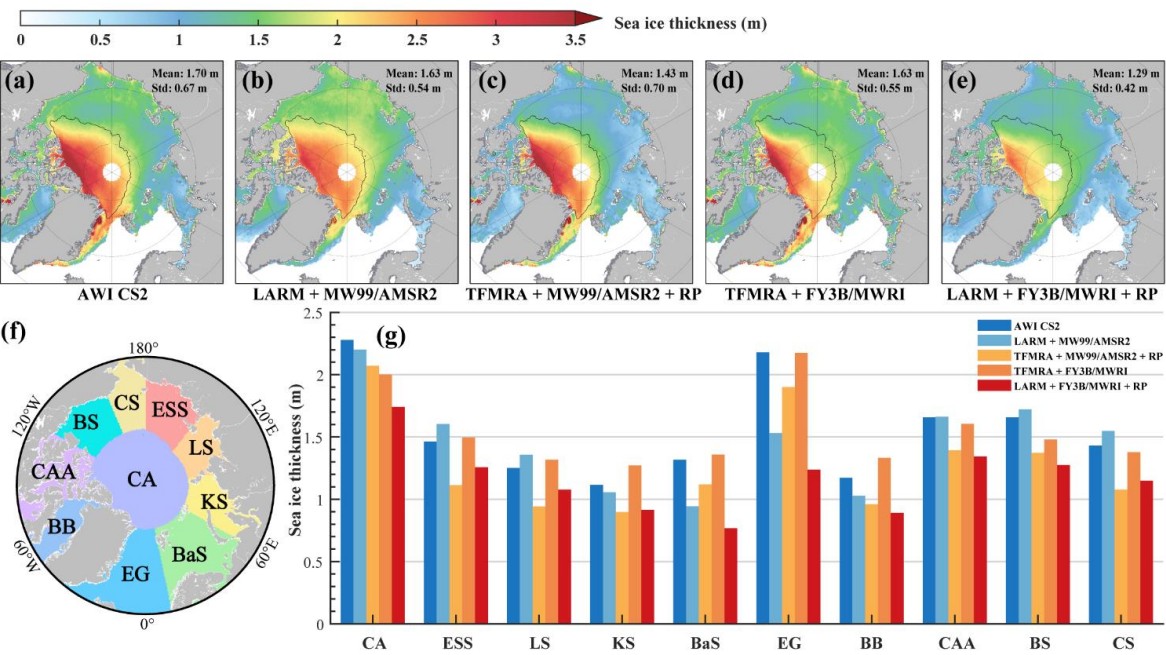

**Fig 7.** Multi-year (2013–2018) mean spatial distribution of Arctic sea ice thickness in the case of (a) AWI CS2, (b) LARM + MW99/AMSR2, (c) TFMRA + MW99/AMSR2 + RP, (d) TFMRA + FY3B/MWRI and (e) LARM + FY3B/MWRI + RP. The black line indicates the boundary between first-year ice and multi-year ice from the ice-type product of the Ocean and Sea Ice Satellite Application Facility (OSI





SAF). (f) describes the sub-regions of the Arctic Ocean, including the Central Arctic (CA), East Siberian Sea (ESS), Laptev Sea (LS), Kara
Sea (KS), Barents Sea (BS), East Greenland (EG), Baffin Bay (BB), Canadian Arctic Archipelago (CAA), Beaufort Sea (BS) and Chukchi
Sea (CS). (g) shows the histogram of the mean sea ice thickness of the cases for each sub-region.

## 4.2 Monthly and seasonal variabilities

To quantify the differences between the optimization cases and the AWI CS2, we investigated the multi-year monthly
(October–April) and seasonal (fall–spring) sea ice thickness (Figs. 8-10). In general, all the optimization cases showed a
370 consistent pattern of having a negative sea ice thickness difference in the MYI region. This negative difference increased from
fall (October–November) to winter (December–February) and spring (March–April). This means that the optimization cases
derived smaller sea ice thickness of MYI than AWI CS2. Significantly, this difference appeared during the period of sea ice
formation. However, the monthly and seasonal differences in sea ice thickness in FYI are not in agreement in the different
optimization cases.

The LARM + MW99/AMSR2 showed positive differences (~0.1–0.3 m) in most regions of FYI in the fall (Fig 8). In
winter and spring, the occurrence of positive differences expanded to the coastal regions and the positive values increased to
~0.3–0.5 m (Figs. 9-10). The probability density function (PDF) of LARM + MW99/AMSR2 shows the peak is always in the
positive sea ice thickness difference.

Since the consideration of radar penetration produces a reduced radar freeboard, the TFMRA + MW99/AMSR2 + RP
showed negative differences in the entire region from October to April (Figs. 8–10). Although the snow depth was generally
greater in MYI than in FYI, the effect of a smaller radar penetration rate in FYI can result in reduced sea ice thickness. Due to
increasing snow depth, the TFMRA + MW99/AMSR2 + RP showed negative differences in FYI that decreased gradually from
~−0.2 m in the fall to ~−0.5 m in the spring. The PDF of LARM + MW99/AMSR2 showed that the differences were
concentrated at one peak in the fall but had a bimodal distribution in winter and spring.

Although TFMRA + FY3B/MWRI also showed positive differences in some of FYI, the distribution of monthly differences
is different from those shown by LARM + MW99/AMSR2. In the fall, the largest positive differences were mainly in the
Kara-Barents Seas and Baffin Bay, with maxima up to 0.7 m. In winter and spring, the extent of these positive differences
shrank and the most positive values decreased to less than 0.4 m. The PDF of LARM + MW99/AMSR2 showed a bimodal
distribution in winter with one peak in the positive direction and the other in the negative direction (Fig. 9).

In contrast to the LARM + MW99/AMSR2 and TFMRA + FY3B/MWRI, the LARM + FY3B/MWRI + RP had negative
differences in most regions of FYI. The negative differences were largest in spring, with a mean value of ~−0.4 m (Fig. 10).
The mean sea ice thickness of LARM + FY3B/MWRI + RP had the largest negative value of all the optimization cases, and
the peak of PDF was always negative.



**Fig 8.** Multi-year monthly sea ice thickness difference (optimization case − AWI CS2) in fall (October and November). The upper figures represent the spatial distribution of sea ice thickness differences between the optimization cases and AWI CS2. The black line indicates the boundary between first-year ice and multi-year ice. The lower figures represent the probability density of sea ice thickness differences of the optimization cases. The green, blue, red, and yellow lines represent LARM + MW99/AMSR2, TFMRA + MW99/AMSR2 + RP, TFMRA + FY3B/MWRI, and LARM + FY3B/MWRI + RP, respectively.

**Fig 9.** Multi-year monthly sea ice thickness difference (optimization case − AWI CS2) in winter (December, January, and February). The upper figures represent the spatial distribution of sea ice thickness differences between the optimization cases and AWI CS2. The black line indicates the boundary between first-year ice and multi-year ice. The lower figures represent the probability density of the sea ice thickness differences of the optimization cases. The green, blue, red, and yellow lines represent LARM + MW99/AMSR2, TFMRA + MW99/AMSR2 + RP, TFMRA + FY3B/MWRI, and LARM + FY3B/MWRI + RP, respectively.

**Fig 10.** Multi-year monthly sea ice thickness difference (optimization case − AWI CS2) in spring (March and April). The upper figures represent the spatial distribution of sea ice thickness differences between the optimization cases and AWI CS2. The black line indicates the boundary between first-year ice and multi-year ice. The lower figures represent the probability density of the sea ice thickness differences of the optimization cases. The green, blue, red, and yellow lines represent LARM + MW99/AMSR2, TFMRA + MW99/AMSR2 + RP, TFMRA + FY3B/MWRI, and LARM + FY3B/MWRI + RP, respectively.



### 4.3 Variation trends

The correlations between the optimization cases and AWI CS2 for 2013–2018 showed that the improved sea ice thickness data
       has the same general variation patterns as the original AWI CS2. The correlations of all data grids were significant ($P < 0.05$).
       With respect to spatial distribution, the TFMRA + MW99/AMSR2 + RP had the highest correlation with the AWI CS2 with
       all correlation coefficients larger than 0.8 (Fig. 11). The TFMRA + FY3B/MWRI distribution of correlation coefficients was
       similar to those of TFMRA + MW99/AMSR2 + RP, but had smaller correlation coefficients of 0.5–08 in the central Arctic.

Since the LARM has a minimum ice thickness retrieval limit of ~0.25 m, which is different from the ~0.5 m of TFMRA (Landy
       et al., 2020), the cases that included the improved retracking algorithm (LARM + MW99/AMSR2, LARM + FY3B/MWRI +
       RP) showed a lower correlation with AWI CS2 in regions with thin sea ice. With respect to temporal variation, all the
       optimization cases had the largest correlation coefficients in October and there was a decreasing correlation trend from fall to
       spring (Fig. 12). The highest correlation was with TFMRA + MW99/AMSR2 + RP, which had coefficients all larger than 0.9.

The lowest correlation was with LARM + FY3B/MWRI + RP, which had coefficients all less than 0.8.

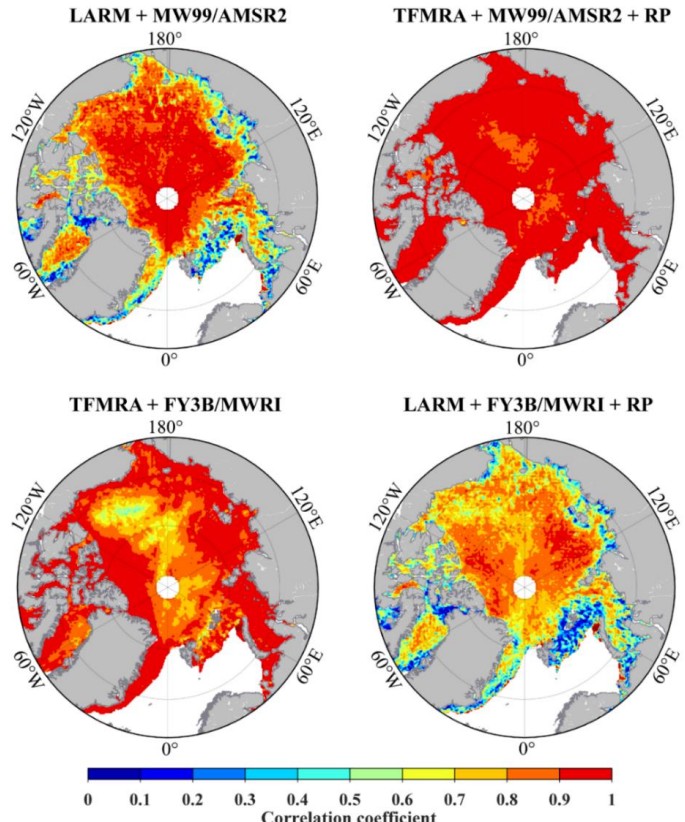

**Fig 11.** Spatial distribution of correlation coefficients ($P < 0.05$) between the optimization cases and AWI CS2 during 2013–2018.

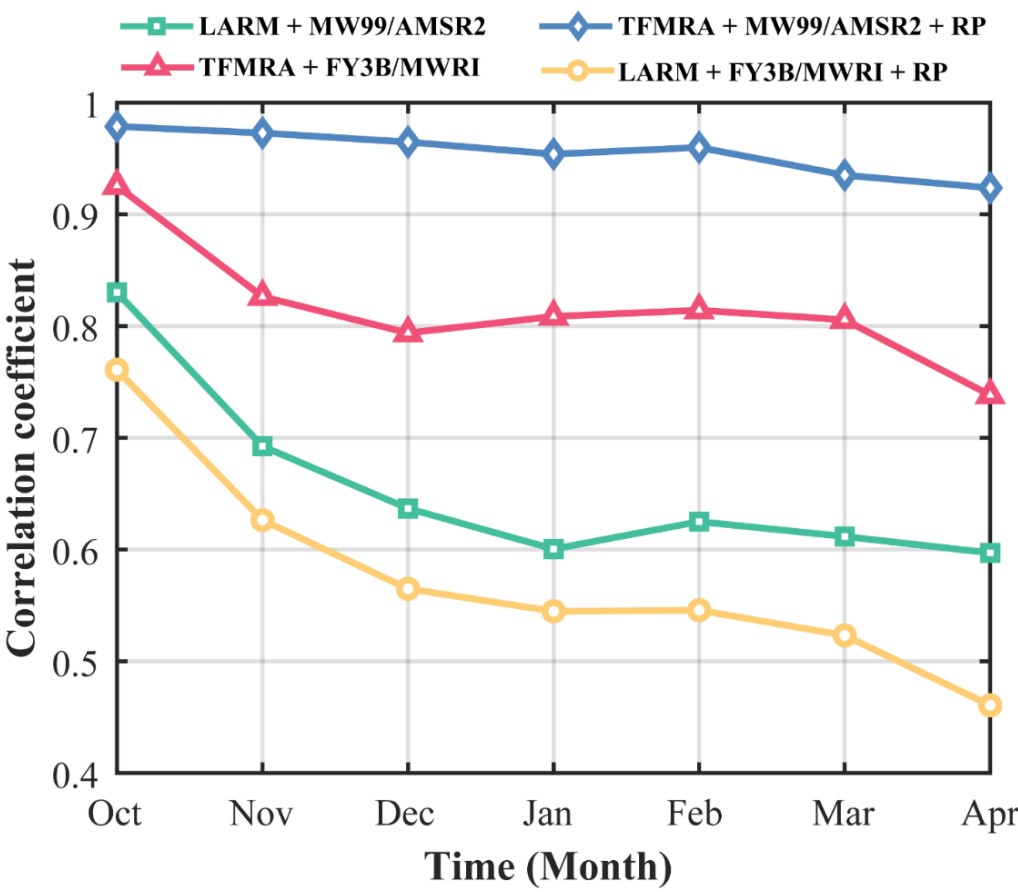

**Fig 12.** Temporal variation of multi-year (2013–2018) monthly correlation coefficients between the optimization cases and AWI CS2 from
October to April.

In the case of AWI CS2, the significant variation trends occurred mainly along the northern coastal regions of the Canadian
Arctic Archipelago and Greenland, in some coastal regions of the Eastern Siberia Sea and Laptev Sea, and some regions of
the central Arctic (Fig. 13). Compared with the variation trend of AWI CS2, all the optimization cases captured the decreasing
trend in the coastal regions of the Canadian Arctic Archipelago and Greenland. The TFMRA + FY3B/MWRI and TFMRA +
MW99/AMSR2 + RP derived a more rapid decreasing trend of −0.17 and −0.15 m yr$^{-1}$ and the LARM + MW99/AMSR2
showed a slightly decreasing trend of −0.10 m yr$^{-1}$. The optimization cases also captured an increasing trend with AWI CS2
in some coastal regions of the Eastern Siberia Sea and the Laptev Sea. The LARM + MW99/AMSR2 and LARM +
FY3B/MWRI + RP clearly exhibited a small increasing trend. However, in the central Arctic, only the TFMRA +
MW99/AMSR2 + RP showed an increasing trend similar to that seen with the AWI CS2. The cases that included the improved
retracking algorithm and new snow depth suggest that the increasing trends of the original AWI CS2 in some regions of the
central Arctic may have been an overestimation.



**Fig 13.** Linear trends of sea ice thickness for the AWI CS2 and the optimization cases from 2013 to 2018. Stippling indicates the statistical
significance of the regression slopes at the 95 % confidence level.

## 5 Discussion

### 5.1 Sensitivity of radar penetration rates calculation

In this study, the effect of corrected radar penetration rates on improving the retrieval of sea ice thickness was verified by
validation. The definition of radar penetration factors plays an important role in sea ice thickness retrieval. Equation (8)
indicates that the accuracy of the radar penetration factor is dependent on total freeboard, snow depth, radar freeboard, and





snow density. In some cases, with unknown total freeboard, such as the in situ observation of IMB, the sea ice thickness and snow depth can be converted to total freeboard through Eq. (9), and the sea ice density becomes another influencing factor.

The radar penetration rates from various in situ observations show the noticeable sensitivity in this study (Fig. 3). In the definition of radar penetration factors, we followed the original AWI CS2 and used constant snow density and sea ice density.
We further explored the impact of snow density and sea ice density on estimating the radar penetration rates. We selected a typical snow density range (270–400 kg m$^{-3}$) (Warren et al., 1999) and sea ice density range (880–940 kg m$^{-3}$) (Alexandrov et al., 2010) to analyze the sensitivity of the radar penetration rates calculation. In the snow density sensitivity experiments, we used OIB Quick Look data from 2013, which has data on total freeboard. In the sea ice density sensitivity experiments, since we needed to use data that does not include total freeboard, we used in situ observation of IMB during 2010–2016.

The sensitivity experiments revealed that sea ice density affects the radar penetration rate calculation more than snow density. In the experiments using OIB Quick Look, the radar penetration rate decreased from 0.85 to 0.78 with an increase in snow density from 270 to 400 kg m$^{-3}$. The radar penetration rate was decreased by only ~8 % when the snow density was increased by 48 % (Fig. 14). Similarly, in the experiments with IMB with a fixed sea ice density, variations in snow density had only a slight effect on the radar penetration rate calculation although the decreasing trend was relatively larger than in the
sensitivity experiments with OIB Quick Look. The radar penetration rate decreased by only ~14–17 % when the snow density was increased by 48 %. However, when the snow density was fixed, the variation of sea ice density had a remarkable effect on the radar penetration rate calculation (Fig. 14). The radar penetration rate was decreased by ~30 % when the sea ice density was increased by 7 % from 880 to 940 kg m$^{-3}$. Therefore, to obtain more accurate radar penetration rates, further investigation of the spatiotemporal variabilities of Arctic snow density and sea ice density is necessary.

In this study, the radar penetration rates were given according to the ice type (FYI and MYI). In fact, the variation of radar penetration rates with time is another consideration. Here, we combined all in situ observations to recalculate the monthly radar penetration rates from October to April. We do not distinguish between sea ice types due to the relatively few sample points of MYI and FYI for each month.

The results showed that the monthly radar penetration rate varied with time (Fig. 15). The radar penetration rates had a range
of 0.79–0.96, with a minimum in December and a maximum in April. Since only IMB had data from October to February, the monthly radar penetration rate for this period was not representative. In addition, we used a single in situ dataset in March and April to get the radar penetration rate and found that the radar penetration rates had large differences. In March, the minimum and maximum radar penetration rate was calculated using IMB and OIB L4, and the difference between them was 0.15. In April, the minimum and maximum radar penetration rate was calculated using AWI IceBird and IMB, and the difference
between them reached 0.16. This suggests that the monthly radar penetration rate is very sensitive to the dataset used. Therefore, if radar penetration rates vary over time, sufficient observational data for each month is essential and the error of observed data needs to be controlled.



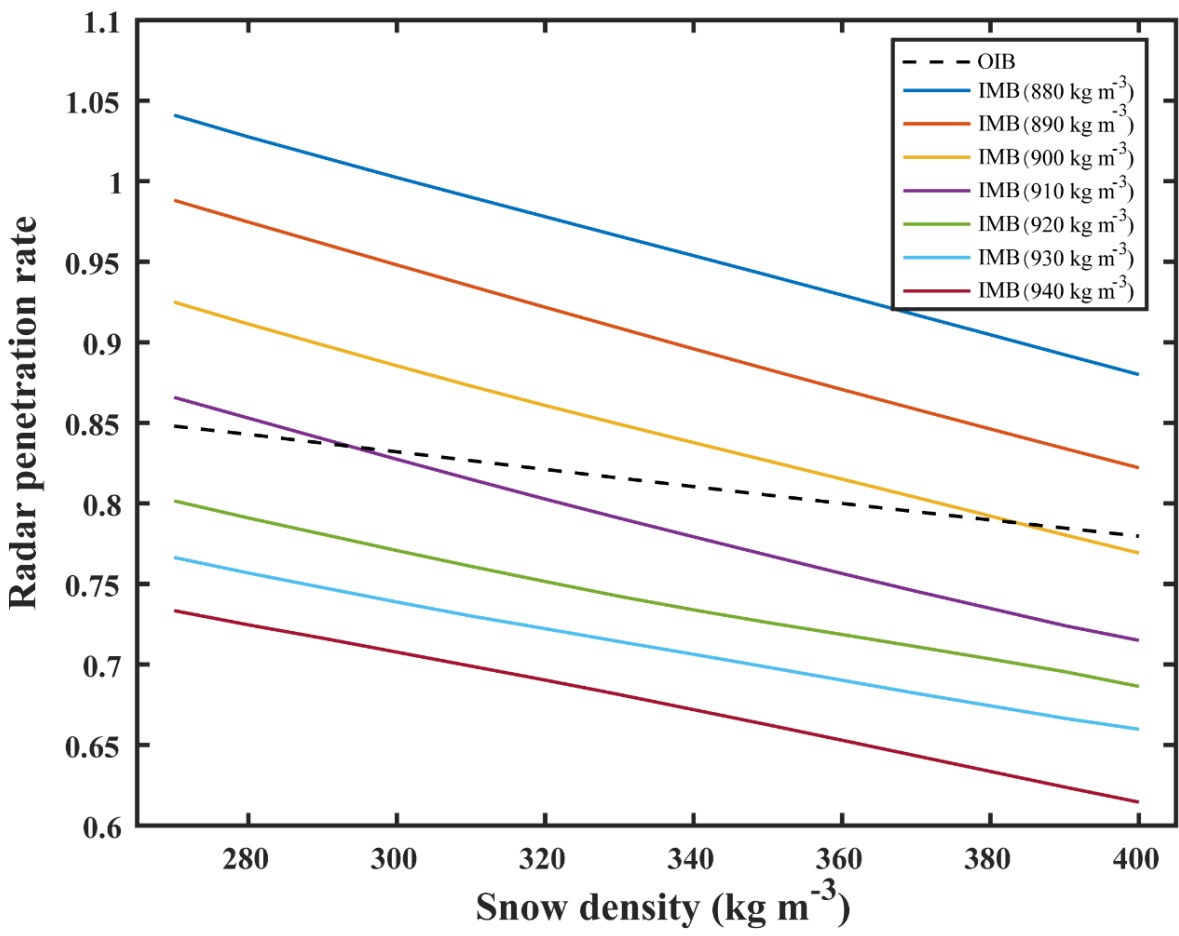

**Fig 14.** Sensitivity of radar penetration rate based on snow density and sea ice density. The black dashed line indicates the radar penetration rates calculated from OIB Quick Look, which is only related to snow density. The solid lines indicate the radar penetration rates calculated from IMB, which is related to both snow density and sea ice density.



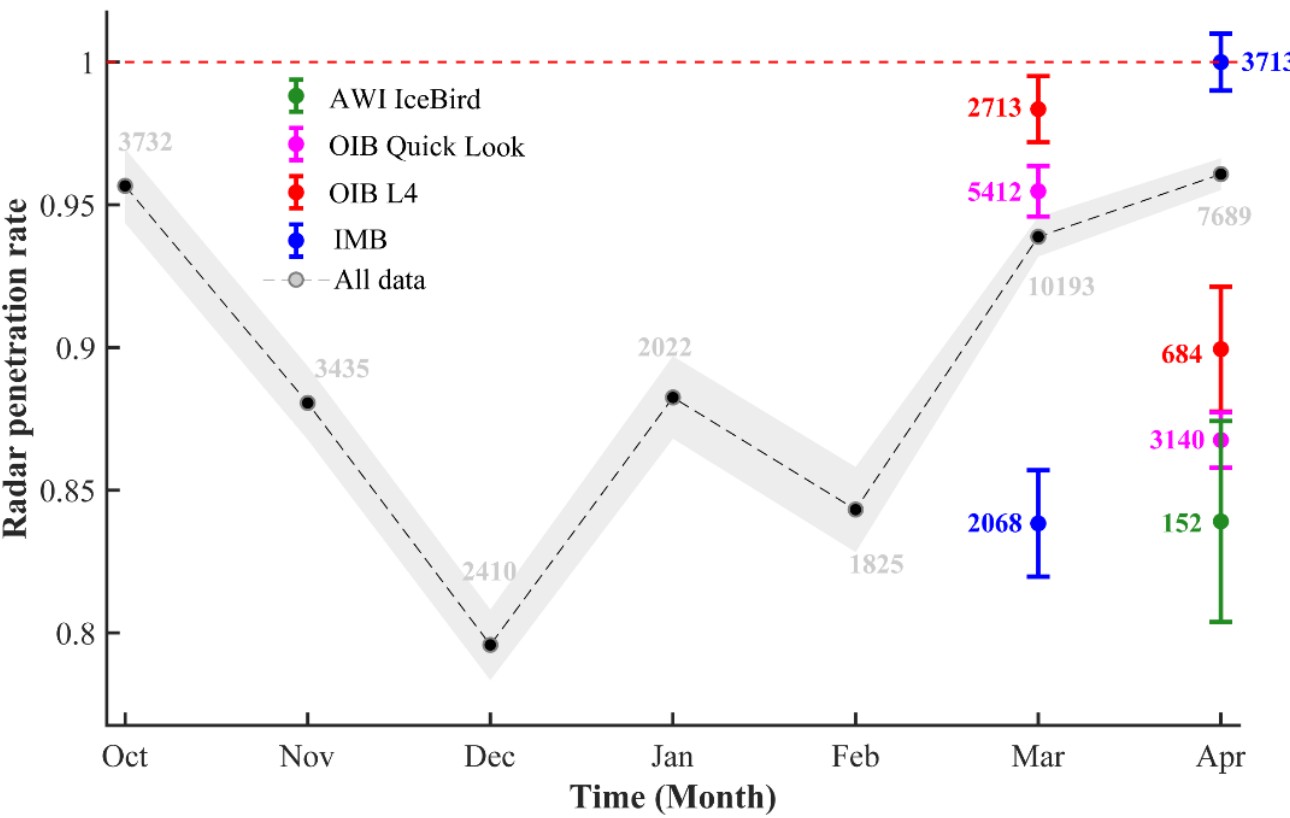

**Fig 15.** Monthly radar penetration rates from October to April. The black scatters indicate the radar penetration rates calculated from all
collected in situ observations, and the shading indicates the uncertainty. The green, purple, red, and blue dots and error bars indicate the
radar penetration rate and its uncertainty calculated from AWI IceBird, OIB Quick Look, OIB L4, and IMB, respectively. The grey numbers
marked in the figure indicate the sample points used to calculate the radar penetration rates.

**5.2 Sensitivity of snow depth to sea ice thickness**

The effect of new snow depth on improving the retrieval of sea ice thickness has also been verified in this study. The selection
of snow depth data is a key factor in improving the sea ice thickness. Currently, relatively complete satellite and in situ
observations of snow depth are few. In this study, we collected three kinds of numerical snow depth data including NESOSIM,
SnowModel-LG, and TOPAZ4. The applicability of different snow depth datasets for improving the retrieval of sea ice
thickness with AWI CS2 was evaluated through validation with OIB Quick Look, CryoVEx, IMB, and AWI IceBird.

The assessment results indicated that the satellite data of FY3B/MWRI and the numerical data of TOPAZ4 were good
choices for snow depth. The sea ice thickness derived from the snow depth of FY3B/MWRI showed the smallest bias of 0.14
m with OIB Quick Look, 0.04 m with CryoVEx, and –0.12 m with AWI IceBird. FY3B/MWRI had the largest correlation
coefficients of 0.64 with OIB Quick Look and 0.42 with AWI IceBird (Fig. 16). In addition, FY3B/MWRI had the most



significant reduction of the RMSE in the validation with OIB Quick Look, CryoVEx and AWI IceBird (Fig. 16). TOPAZ4 had the second smallest bias (0.42 m) in the validation with OIB Quick Look and second largest correlation coefficients (0.77

and 0.37) with CryoVEx and AWI IceBird (Fig. 16). TOPAZ4 also reduced the RMSE of sea ice thickness in the validation with OIB Quick Look, CryoVEx and AWI IceBird (Fig. 17). The sea ice thickness derived from NESOSIM and SnowModel-LG had smaller correlation coefficients than the AWI CS2 in the OIB Quick Look validation. In addition, the sea ice thickness derived from NESOSIM had a larger bias and smaller correlation coefficients than AWI CS2 in the validation of AWI IceBird. The sea ice thickness derived from SnowModel-LG did not reduce the RMSE in all of the in situ observations and the sea ice

thickness derived from NESOSIM reduced the RMSE only in the validation of CryoVEx.

These comparison results do not reveal the accuracy of snow depth datasets. Instead, most of these data have been evaluated in detail (Li et al., 2021; Petty et al., 2018; Stroeve et al., 2020). These results provide some reasonable suggestions for the selection of snow depth to improve sea ice thickness when other factors are consistent with the original AWI CS2.

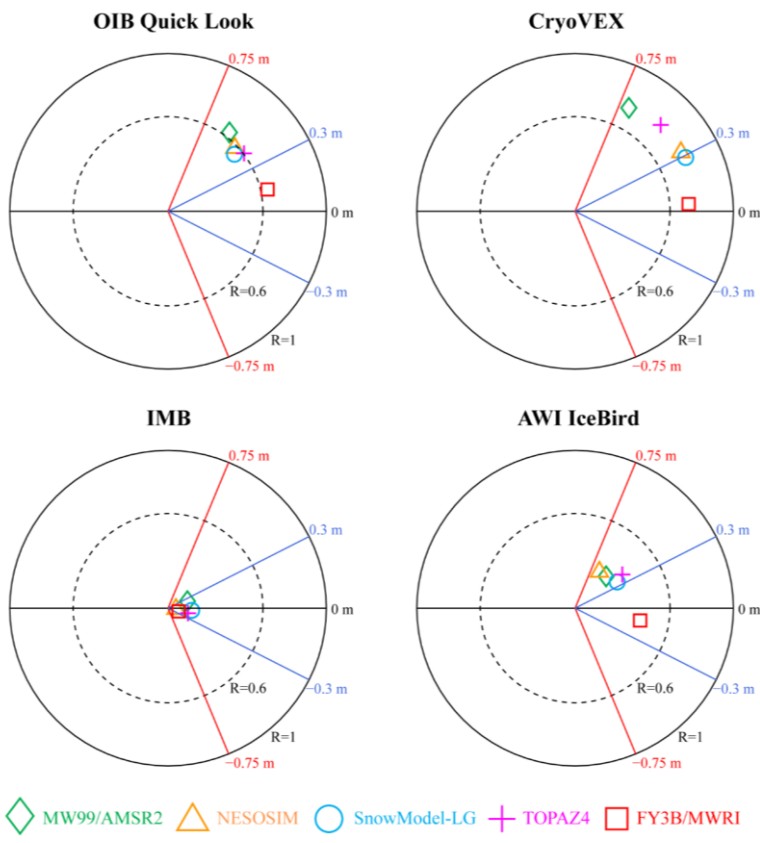

**Fig 16.** Taylor diagrams showing the validation results of sea ice thickness derived from the snow depth of MW99/AMSR2, NESOSIM, SnowModel-LG, TOPAZ4, and FY3B/MWRI with the in situ observations. The radial angle of ±π indicates the bias of ±2 m. The distance from the origin indicates the correlation coefficient.




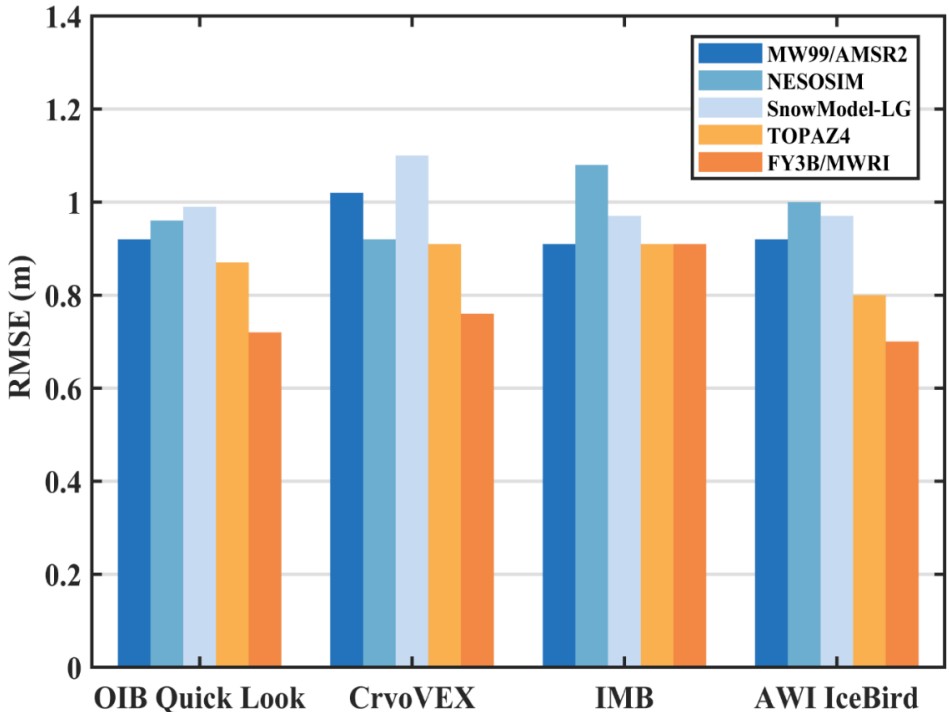

**Fig 17.** Root mean square error of sea ice thickness derived from the snow depth of MW99/AMSR2, NESOSIM, SnowModel-LG, TOPAZ4,
and FY3B/MWRI with the in situ observations.

## 5.3 Impact of sea ice density on sea ice thickness retrieval

In this study, the selection of sea ice density was consistent with the AWI CS2. However, some results suggested that the
impact of sea ice density on sea ice thickness retrieval is important (Kwok and Cunningham, 2015; Zygmuntowska et al.,
2014). Jutila et al. (2021) obtained updated sea ice densities based on airborne measurements in 2017 and 2019 (FYI = 925 kg
$m^{-3}$ and MYI = 902 kg $m^{-3}$). These densities were larger than the densities used in the AWI CS2 (FYI = 917 kg $m^{-3}$ and MYI
= 882 kg $m^{-3}$). To investigate the impact of sea ice density on improving the retrieval of sea ice thickness, we replaced our sea
ice densities with those from Jutila et al. (2021) and revalidated the sea ice thickness of the four optimization cases.

The revalidation results suggest that the impact of different sea ice densities on the sea ice thickness retrieval is
nonnegligible. With the updated sea ice densities, all the optimization cases still improved the retrieval of sea ice thickness
with AWI CS2, but the magnitude of the improvement varied with the different cases. In the optimization cases that included
the improved retracking algorithm, corrected radar penetration rate, and new snow depth, the improvement in sea ice thickness
retrieval using the original sea ice densities was slightly better than that using updated sea ice densities. The difference in the
RMSE between the cases using original and updated sea ice densities had a range of 0.03–0.39 m (Table 3). In the combined
optimization case (LARM + FY3B/MWRI + RP), using updated sea ice densities reduced the RMSE to 0.05–0.16 m. The
correlation coefficients between the cases using original and updated sea ice densities were similar.



Since the RMSE of AWI CS2 increased with all in situ observations based on updated sea ice densities, it is not appropriate to individually replace the original sea ice densities. The replacement of sea ice densities was combined with the optimization of the retracking algorithm, radar penetration rate, and snow depth. In addition, due to the different performances of sea ice thickness improvement in four optimization cases, it is difficult to determine which sea ice density scheme is better; however, both of them are applicable.

**Table 3.** Validations of sea ice thickness or total thickness with in situ observations based on the updated and original sea ice densities. The values in the brackets are the results using original sea ice densities.

| Retrieval scheme | IMB | | CryoVEX | | AWI IceBird | |
|---|---|---|---|---|---|---|
| | R | RMSE (m) | R | RMSE (m) | R | RMSE (m) |
| AWI CS2 | 0.14 (0.13) | 1.09 (0.91) | 0.75 (0.74) | 1.44 (1.02) | 0.25 (0.28) | 1.11 (0.92) |
| LARM + MW99/AMSR2 | 0.28 (0.28) | 0.81 (0.64) | 0.72 (0.69) | 1.00 (0.77) | 0.16 (0.19) | 0.98 (0.79) |
| TFMRA + MW99/AMSR2 + RP | 0.11 (0.10) | 1.02 (0.89) | 0.78 (0.78) | 1.26 (0.87) | 0.15 (0.19) | 0.94 (0.79) |
| TFMRA + FY3B/MWRI | 0.08 (0.07) | 0.96 (0.91) | 0.73 (0.72) | 0.95 (0.76) | 0.41 (0.42) | 0.73 (0.70) |
| LARM + FY3B/MWRI + RP | 0.16 (0.15) | 0.64 (0.69) | 0.79 (0.77) | 0.61 (0.77) | 0.31 (0.33) | 0.72 (0.81) |

## 6 Conclusions

It was found that compared with other sea ice parameters such as sea ice concentration, sea ice thickness was relatively difficult to estimate due to uncertainty in sea ice freeboard and snow depth. In the AWI CS2 sea ice thickness product, the retrieval errors were caused by sea ice surface roughness, snow backscatter, and snow depth on sea ice. In this study, three optimizations of an improved retracking algorithm (LARM), the corrected radar penetration rate (modified AR15), and the new snow depth (FY3B/MWRI) were used for the first time to improve the three kinds of uncertainty mentioned above. To quantify the improvement in sea ice thickness retrieval, three individual optimization cases of improved retracking algorithm (LARM + MW99/AMSR2), corrected radar penetration rate (TFMRA + MW99/AMSR2 + RP), new snow depth (TFMRA + FY3B/MWRI) and one combined optimization case (LARM + FY3B/MWRI + RP) were created and validated with in situ observations from OIB Quick Look, IMB, CryoVEX, and AWI IceBird.

In improving the retrieval of sea ice thickness, the sea ice freeboard was recalculated using an improved retracking algorithm and corrected radar penetration rates. The corrected radar penetration rates were 0.77 for FYI, 0.96 for MYI, and 0.91 for all ice types based on all in situ observations. Compared with the original sea ice freeboard of AWI CS2, the sea ice freeboard in the optimization cases was generally 0.01–0.06 m smaller. The new snow depth of FY3B/MWRI was about 0.03 m smaller than the MW99/AMSR2 used in the AWI CS2.





In the validation of sea ice freeboard, sea ice thickness, and total thickness, the results showed that all the optimization cases
had the ability to improve the retrieval of the sea ice freeboard, sea ice thickness, and total thickness with similar degrees of
correlation with the in situ observations. In the validation of sea ice freeboard, the optimization cases reduced the RMSE by
0.51–3.27 cm. The LARM + FY3B/MWRI + RP produced the largest reduction. In the validation of sea ice thickness, the
optimization cases improved the retrieval values with a reduction of the RMSE up to 0.23 and 0.27 m (25.0 % and 29.7 %)
compared with the in situ observations of OIB Quick Look and IMB. The LARM + FY3B/MWRI + RP and LARM +
MW99/AMSR2 showed the most remarkable improvement with the OIB Quick Look and IMB. In the validation of total
thickness, the optimization cases improved the RMSE up to 0.26 and 0.22 m (25.5 % and 23.9 %) compared with in situ
observations of CryoVEX and AWI IceBird. The TFMRA + FY3B/MWRI had the largest improvement with these two
observational datasets.

  The spatiotemporal patterns of sea ice thickness based on improved data showed noticeably different from the original
findings from the AWI CS2. The spatial patterns of sea ice thickness indicated that all the optimization cases maintain the
major patterns of sea ice thickness distribution, but in general, showed relatively smaller sea ice thickness. In some sub-regions
of the Arctic Ocean, the optimization cases with improved the retracking algorithm (LARM + MW99/AMSR2) and new snow
depth (TFMRA + FY3B/MWRI) produced greater sea ice thickness. Compared with the AWI CS2, all the optimization cases
had smaller sea ice thickness in the MYI region. The differences became increasingly evident from fall to spring, which is the
period of sea ice formation. The sea ice thickness differences in the FYI region varied by optimization case. Although the
optimization cases showed similar temporal variations in sea ice thickness and had large correlation coefficients with the AWI
CS2, the differences in the variation trends between the optimization cases and AWI CS2 were significant in the coastal regions
of the Canadian Arctic Archipelago, Greenland, Eastern Siberia Sea, and Laptev Sea and some in regions of the central Arctic.

  The sensitivity experiments exploring the impact of snow density and sea ice density on estimating the radar penetration
rates revealed that the radar penetration rate calculation is more sensitive to sea ice density than to snow density. The radar
penetration rate decreased by ~30 % when the sea ice density increased by 7 % from 880 to 940 kg m$^{-3}$. The sensitivity
experiments investigating the effect of different snow depths on improving the retrieval of sea ice thickness showed that the
snow depth of FY3B/MWRI and TOPAZ4 can be good choices for the improvement of sea ice thickness retrieval. Compared
with the sea ice density scheme of AWI CS2 (FYI = 917 kg m$^{-3}$ and MYI = 882 kg m$^{-3}$), the updated sea ice density scheme
(FYI = 925 kg m$^{-3}$ and MYI = 902 kg m$^{-3}$) used together with the comprehensive optimizations can improve the retrieval of
sea ice thickness from AWI CS2. However, the magnitude of the improvement varied with the different optimization cases.
The comprehensive comparisons suggested that two kinds of sea ice densities are applicable.

  This study highlighted the successful optimizations of a retracking algorithm, radar penetration, and snow depth to improve
the retrieval of sea ice thickness derived from CryoSat-2. The improvement strategies proposed in this study could be
considered in future sea ice thickness retrieval processes of AWI CS2. The assessment results of sea ice thickness could help
to further understand the uncertainties of satellite monitoring and variations of sea ice volume. Additional efforts to reconcile
the altimetry missions of sea ice thickness such as CryoSat-2, ICESat-2, and other upcoming satellites are required.



*Data availability.* The sea ice thickness and radar freeboard of CryoSat-2 derived by AWI is available at
ftp://ftp.awi.de/sea_ice/product/cryosat2. The radar freeboard derived from the Lognormal Altimeter Retracker Model (LARM)
is available at https://data.bas.ac.uk/full-record.php?id=GB/NERC/BAS/PDC/01257. The snow depth of FY3B/MWRI is
through the communication with the authors. The snow depth of MW99/AMSR2 is available at
ftp://ftp.awi.de/sea_ice/auxiliary/snow_on_sea_ice/w99_amsr2_merge. The snow depth of NESOSIM is available at
https://zenodo.org/record/5164314#.Yel_N3pBw2x. The snow depth of SnowModel-LG is available at
ftp://ftp.cira.colostate.edu/ftp/Liston/SnowModel_LG_1980-2018/. The snow depth of TOPAZ4 is available at
https://data.marine.copernicus.eu/product/ARCTIC_MULTIYEAR_PHY_002_003. The observed dataset of OIB L4 and OIB
Quick Look are available at https://nsidc.org/data/IDCSI4/versions/1 and https://nsidc.org/data/NSIDC-0708/versions/1. The
observed dataset of IMB is available at http://imb-crrel-dartmouth.org. The observed dataset of CryoVEx is available at
https://earth.esa.int/eogateway/campaigns/cryovex-aem. The observed dataset of AWI IceBird is available at
https://doi.org/10.1594/PANGAEA.932668 and https://doi.org/10.1594/PANGAEA.927448.

*Author contributions.* YZ and YZ conceptualised the study, developed the methodology and carried out the main analysis. YZ
and YZ prepared the manuscript under the supervision of CC and RCB and all authors contributed to revising the manuscript.
CC, DX, RCB and WS contributed to the interpretation of the results. LL developed the snow depth of FY3B/MWRI.

*Competing interests.* The authors declare no competing interests.

*Acknowledgements.* We are very thankful to Robert Ricker (rori@norceresearch.no) for his positive comments to our work
and providing us the constructive guidance which helped to improve and clarify the paper. We are also grateful to editor Michel
Tsamados and the team at The Cryosphere for their support and processing of our paper. This work is sponsored by the National
Key Research and Development Program of China (No.2019YFA0607001), Natural Science Foundation of Shanghai
(No.22ZR1427400), National Natural Science Foundation of China (No.42130402 and No.41706210), and the Innovation
Group Project of Southern Marine Science and Engineering Guangdong Laboratory (Zhuhai) (No.311022006).

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
