# Peer review of "Improving Arctic sea ice thickness retrieved from CryoSat-2: A comprehensive optimization of a retracking algorithm, radar penetration rate, and snow depth"

_The Cryosphere, 2023_

## Referee Comment (RC1)

**Synopsis**

This work seeks to address several open research questions surrounding the amount of snow on sea ice, the ability of the CryoSat-2's radar waves to penetrate snow, and radar waveform interpretation. Reducing one of these uncertainties may move the subsequently retrieved value further away from "the truth" due to the presence of compensating biases, so it makes sense to consider several sources of uncertainty at once and then optimise. This work is therefore well motivated.

In this submission the authors have interchanged several aspects of the radar-freeboard to sea-ice-thickness processing chain. In particular, they have used two different "off the shelf" radar freeboard products (AWI and Bristol/UIT), several snow products (AMSR/W99, SnowModel-LG, NESOSIM, TOPAZ4, FY3B/MWRI), and have compared the assumption of full radar penetration of the assumed snowpack to an ice-type dependent radar penetration factor. These radar penetration factors were derived from comparison to several separate (but often not independent) data sources such as airborne estimates of sea ice freeboard/thickness and drifting ice-mass-balance buoys (Fig. 3). Having made these various interchanges, the authors appear to compare their new SIT data to similar evaluation data as before, which seems like a conflict between training and testing (Table 2; Fig. 5).

To be honest, I found it difficult to identify any clear conceptual or practical advances in this work. I was also left somewhat doubting whether the claims in the abstract and conclusions were supported by the results. It seems to me that rather than a "comprehensive optimisation", this work represents several experimental recombinations of different existing components in the processing chain. Furthermore, the authors' recombinations are potentially improving one metric of skill (e.g. RMSE) at the expense of another metric (e.g. correlation R value). I also note that not all combination possibilities were explored.

I was also concerned about the treatment of OIB QuickLook data as "in situ validation" (Table 2; Figure 5), since that particular data set has several known issues and is of course an airborne remote sensing product (not in-situ). Similarly, AWI IceBird and CryoVEX airborne data are also referred to and treated as "in situ validation", but are of course also the results of airborne remote sensing with their own biases and uncertainties. Particular to CryoVEX, it is **not** appropriate to use Ku-band ASIRAS data processed with the assumption of full radar penetration to then work out CS2 penetration factors.

Relatedly, the issue of how representative an ice-mass-balance buoy's measurement is of the ice sampled by (a) a CryoSat-2 SAR footprint, or (b) the ice in a 25x25km grid cell such as those analysed here, was mentioned (Sect. 2.4) but not meaningfully tackled. These issues strongly affect the suitability of IMBs for radar-altimetry evaluation. More broadly, the extent to which the authors may be optimising towards an uncertain or even biased truth is not discussed.

Finally, it was not possible for me to reproduce this analysis since it relies substantially on an unpublished snow dataset from the FY3B radar altimeter (which one coauthor has previously led a publication on). Since the manuscript relies so heavily on publicly available data, I thought it was a shame not to act in this spirit. The reason for the data not being made available was not given (despite an explanation being required by the Copernicus data policy). On a related note, no code was made available with this manuscript for review. These factors will limit the impact of any publication, and also limit my confidence as a reviewer.

It is therefore my recommendation that this manuscript should be rejected, and a similar analysis reperformed and resubmitted in service of a more carefully constructed research question. I appreciate this might be received as an overly negative recommendation, but I believe our community has a long way to go until we can meaningfully conduct the comprehensive optimisation marketed in this manuscript. Such an exercise would require reprocessing of several key datasets by many different research groups for a truly "apples to apples" comparison, not to mention a deep new look at the sampling biases, systematic uncertainties and independence of airborne and in-situ evaluation data.

**Major Concerns**

**Operation IceBridge snow depth, freeboard and thickness data are not "in-situ validation"**

Firstly (and this is a somewhat trivial point), they are not "in-situ" because they are taken from an aircraft. They are much better described as data products from airborne radar (and sometimes lidar) remote sensing.

Less trivially, because they're radar-based products, they're subject to significant uncertainties involving radar penetration of snow, sea ice roughness, and particularly sidelobes. Much has been written about OIB snow depth retrievals, and the quicklook product that the authors have relied upon in several sections is notoriously poor. In particular, it has relatively poor performance over multiyear ice (King et al., 2015), and the underlying algorithm suffers from a persistent issue of misidentifying range sidelobes from the snow-ice interface as the snow-air interface (Kwok and Maksym, 2014; Kwok and Haas, 2015). Figure 4 of Kwok et al. (2017) illustrates that the GSFC-NK product (which I think corresponds to the OIB QL product?) shows persistently low snow depths. In any case, the figure definitely shows that several very different interpretations of the OIB raw data exist, and therefore any individual snow or ice product cannot represent some kind of "in-situ" truth.

Even in the case where there are no snow-penetration, roughness or sidelobe-related biases, OIB sea ice thickness estimates still rely on exactly the same uncertain conversion of freeboard to thickness as CryoSat-2, so cannot form an independent benchmark for optimisation. For instance, I was hoping for some analysis of how the snow and ice densities used the in OIB hydrostatic conversions compared to those being used in the satellite conversions at hand, but that was glossed over. This type of material must be foundational to any activity that aims to "optimise" satellite retrievals to these OIB products. Otherwise the optimised product be biased in all the ways in which the OIB products are biased. Along these lines, describing OIB thickness products as "observations" is also dubious. The confidence with which you can "observe" SIT even with accurate measurements of snow depth and freeboard is nicely documented by Alexandrov et al. (2010; see numbers in second half of abstract).

**CryoVEX Data**

The authors need to be much more specific and thoughtful about how CryoVEX data on total thickness are generated – it's not enough to say "airborne electromagnetic sensors". If the CryoVEX method is similar to CryoSat-2, it will suffer from similar biases as CryoSat-2 and therefore is unsuitable for independent evaluation.

Furthermore, it appears that the authors have used a very limited subset of CryoVEX data from 2014. Aside from limiting the applicability of their results, my understanding is that SIT for the CryoVEX '14 campaign is from combination of the Ku-band ASIRAS system and the ALS system (Hvidegaard et al., 2015). Given the study by Willatt et al. (2011) on ASIRAS radar penetration through snow on sea ice, I think a lot more consideration needs to be given to the positioning of this data as some kind of objective truth that can be used to optimise satellite data. It's also worth pointing out that if you look at Fig. 9 from Garnier et al. (2021) which analyses CryoVEX 2017 data, it's clear that the difference between the ALS and ASIRAS data is often negative towards the end of the track, further indicating that the "off-the-shelf" approach to CryoVEX ALS/ASIRAS data in this manuscript may need considerable further scrutiny. At an absolute minimum, the authors should understand they cannot get at the CS2 radar penetration factor using Ku-band CryoVEX data that assumes a 100% penetration factor from ASIRAS. When comparing CS2 freeboards to ASIRAS freeboards derived from the assumption of a 100% RP, is it any wonder that we find high CS2 radar penetration factors?

**Ice mass balance buoys**

There are major issues with using single IMBs to validate measurements by radar altimeters such as CryoSat-2. IMBs are generally deployed on level ice, and are of course point measurements. This is in contrast to as CryoSat-2 footprint which is several hundred metres long and several kilometers wide depending on the ice roughness. And of course this is very different to the 25x25km scale on which this manuscript's analysis is conducted. It's also worth considering that IMBs essentially break as soon as the ice

on which they're situated begins to grow dynamically, and dynamic growth can be quite a significant contributor to the thickness in a CS2 footprint and also in a 25 km grid cell. So much more serious consideration needs to be given to how and whether the IMB data correspond to that collected by CryoSat-2, and whether they are suitable for a tuning/optimisation exercise such as this.

**Radar Penetration Factors**

I assume 0.77, 0.96, and 0.91 for the penetration factors were generated from the means of the histograms in Fig. 3A (not the modes, and not the mean of the Gaussian fits) – this should be clarified. More information is also required on how the "all data" histograms are constructed. This is because there are presumably many more data points from some sources than from others: were their contributions weighted for measurement reliability or quantity? It seems like the only reason that FYI penetration is significantly lower than MYI is because of the contribution of IMB data (panel f), but this is probably related to the fact that IMBs have a distinct sampling bias relative to both the other data and their surrounding environment (discussed above). So is it reasonable to allow the derived penetration factor to be so strongly influenced by the IMB contribution? This again goes back to how different sources of evaluation data should be weighed against each other in terms of reliability and quantity.

With the exception of the IMB panel, the histograms generally exhibit a large number of data points with penetration factors >1. This is often >1/3 of the data in a histogram. Since this is unlikely to be physical, it points to other biases existing in the radar altimetry processing chain that need to be accounted for before this manuscript's method will actually work to retrieve the real penetration factor. At the moment, the penetration factor is simply being used as a tuning parameter to make the satellite products match the evaluation data. When deployed in this way the penetration factors are just making up for insufficiencies in the retracking approach/ snow data /hydrostatic conversion, but are also being tuned towards evaluation data that may themselves be biased. This is a totally different concept to what the authors claim to be doing elsewhere in the manuscript, i.e. recovering the penetration characteristics of Ku-band radar waves into snow.

How do the authors justify the width of the uncertainty bars in Figure 15? The shaded bars around the line seem surprisingly thin for a quantity that's eluded the CryoSat community so effectively over the last decade. Could it really be the case that the penetration-factor consistently goes up in January and then back down in Feb by a magnitude considerably larger than the uncertainty bars? What is the spatial range of applicability of these statistics? How are the printed numbers of distinct in-situ observations generated? After all, there are hundreds of thousands of SnowRadar/ASIRAS waveforms and ALS spot heights going into this analysis, So I guess the figures in the thousands correspond to the number of contributing 25x25km grid cells? If more observations go into an aggregated 25x25 km data point, then we should probably weight that data point's reliability as being higher than a grid cell that contains only a few data points. Has this been factored into the uncertainty analysis?

I have several further questions about the radar penetration factor analysis, both concerning how it was done and how it is reported, but will leave it here.

**Radar freeboard data and retracking**

I don't think it's legitimate to claim to have applied a "comprehensive optimisation of an improved retracking algorithm" (from the abstract) when no retracking takes place in the manuscript. This is misleading to the casual reader. A comparison has not been made between retracking algorithms (TFMRA & LARM), but instead between two radar freeboard products (AWI & Bristol/UIT). To imply that differences in the RF products only represents differences in the retracking approach is a risky business. The authors state "although the different classifying waveforms, geophysical corrections, and sea level tie-point interpolation also contribute to a relatively small extent (Landy et al., 2020)." That's not exactly my understanding from reading that paper; Figure S3 shows there definitely are some differences that derive from the "in-house" treatments of AWI vs Bristol/UIT. That's the reason that Landy et al. 2020 emulated the other retrackers with the same processing: to learn about the retrackers in isolation. I think that's the approach that should be taken here if the retrackers are to truly be "optimised" rather than the radar freeboard products.

**Optimisation Metrics**

Firstly the metric referred to as *R* needs some clarification. Is this the Pearson product-moment correlation coefficient? *R* is also often used for the coefficient of determination, which essentially captures the data's deviation from the line y=x (i.e. it also captures the bias, where Pearson does not).

Using the product-moment correlation coefficient has its pitfalls. i.e. with heavy optimisation for Pearson you end up capturing the variance of your evaluation data, but at the cost of increasing your bias. This issue should be handled explicitly. These concepts are at the core of the "bias-variance tradeoff", and allied concepts in optimisation such as overfitting. While RMSE is sensitive to the bias, it also can be large when spread about the y=x line is large and the bias is small. So I think the metrics against which the processing chain is optimised need much more careful consideration when defining the optimisation exercise.

**Some More Minor Things**

The effect of reducing the radar penetration factor from unity to some fractional value is basically to reduce the subsequently calculated ice freeboard and thus the derived thickness. This does the same thing as assuming a deeper snowpack. Since the authors are varying the snow and the RP at the same time, they should grapple with this conflict explicitly: when we observe the sea ice to be "too thick" relative to our evaluation data, we often don't know whether it's because our assumed RP is too high, or whether our assumed snow is too thin. How can we optimise in light of this? One option is to just assume that airborne surveys of snow depth are correct, and use that snow depth to then optimise the RP. The authors haven't done this, but it might be interesting? But it does take a bit of a leap of faith regarding airborne snow depths.

Title: I think we should steer clear of subjective adjectives like "comprehensive" in titles. One could easily argue that this analysis is not comprehensive as it uses a fairly limited set of in-situ sea ice thickness measurements relative to those available, and uses limited subsets of CryoVEX and IceBird data. I recommend removal of this word.

L42: "Recent advances in satellite altimetry began in 2003." – This is a very subjective sentence and I'm not sure what it adds. Envisat began operating in 2002 for instance. What defines "Recent" here?

L68: Lower errors, not minimal.

L140: I think it would be better to cite the official NSIDC source for these data: https://nsidc.org/data/nsidc-0758/versions/1

L509: Perhaps I've missed this, but I don't think the authors have specified whether they're using the ERA5 or MERRA2 run of SnowModel? When comparing SnowModel and NESOSIM it's of course important to make sure they're forced by similar data, or else you're basically just comparing reanalysis precipitation data rather than the models themselves. Same with TOPAZ – I assume this is driven by some ERA-type product?

Fig. 1: There's no reference to the bathymetry in this paper, so I think shading it into this plot confuses things.

Fig. 2: If you label the "Reference ellipsoid" in this figure you should reference/explain it in the text.

L379: Think this sentence needs rewording for clarity.

L447: The authors should discuss their optimised radar penetration factors with reference to Nab et al. (2023), who derived RP factors based on the relationship between radar freeboard and SnowModel-LG depths.

L594: I was disappointed to see that the snow depth data from Li et al. (2021) are not made publicly available, so many of the results presented in this paper are not replicable by either me as a reviewer, or the community at large. As well as limiting my confidence as a reviewer, this will limit the impact of the paper since the supposedly optimised products are not available or reproducible in future. The Copernicus

Publications data policy states "if data are not publicly accessible, a detailed explanation of why this is the case is required", and I can't see why this was ignored. Furthermore, no code was made available with the manuscript, which also limits my confidence in recommending the paper for publication.

---

## Referee Comment (RC2)

**General comments**

The paper aims to address several difficulties encountered in CS-2 (CryoSat-2) data analysis, including waveform retracking, snow depth estimation, and radar penetration. It is noteworthy that the paper summarizes a substantial amount of observational data for validation, compares snow reconstructions, and employs two CS-2 retracking algorithms to generate an ice thickness dataset. However, I share Reviewer 1's opinion that the paper lacks comprehensive optimization. Instead, it leans towards combining algorithms and products in a somewhat selective manner, which is inappropriate for a scientific paper, particularly one focused on datasets or products. Additionally, there is a consistent mixing of sensitivity studies and validation studies, making the paper challenging to read and diminishing its overall credibility. Furthermore, there is a critical need for clarification and correction regarding the definition of radar penetration and potential misunderstandings arising from previous papers.

I acknowledge the usefulness and instructive nature of referring to Armitage et al. (2015) to gain a quick understanding of the priorities concerning the CS-2 radar penetration problem. However, the algorithms employed in the current paper are essentially the same as those presented in Armitage et al. (2015), which calls for innovation and reorganization given the passage of several years. It is important to exercise caution, particularly in two areas: (1) the potential misuse and conflict arising from using training data for radar penetration estimation and validation data for sea ice thickness validation, an issue that Reviewer 1 has already highlighted; and (2) a genuine misunderstanding of how the paper defines the radar penetration factor. The authors seem to suggest that the radar penetration factor depends on the waveform retracking algorithms and snow depth product. However, this may lead to significant misunderstandings since the radar penetration effects are a result of snow and ice scattering, which, in turn, depend on the properties of the upper snow and ice layers, such as wetness, density, and grain size. Therefore, the radar penetration of CS-2 should be based on the properties of the snow and ice itself, rather than solely relying on the waveform algorithm or snow product.

If the authors aim to address the radar freeboard penetration or snow scattering problem, a more appropriate and physically robust approach would be to follow the methodology outlined by Slater et al. (2019), where the penetration depth over Greenland was derived, or refer to the study by Kurtz et al. (2014), which used a model to address the uncertainty of radar penetration. If the authors genuinely seek to reduce bias in a specific radar freeboard estimation, I would suggest the following steps: (1) select an appropriate snow product; (2) choose an appropriate radar freeboard estimation (or retracking algorithm); and (3) based on current in-situ observations, correct the selected radar freeboard bias. In this approach, the fundamental tasks involve selecting the proper snow product and radar freeboard estimation, which have not been adequately addressed in the paper. Additionally, the datasets used for correction and validation purposes lack clarity. Therefore, I strongly recommend that the authors consider changing the term "radar penetration factor" to "radar correction coefficient" to better align with their protocols and enhance the overall structure of the paper.

Based on the aforementioned concerns and the following comments, I suggest rejecting this manuscript. Here are my main comments:

(1) Line 15, '*applying a comprehensive optimization of an improved retracking algorithm, corrected radar penetration rate*'. When reading this line, it gives the impression that the paper will introduce a genuinely improved retracking algorithm, such as better waveform fitting or more reasonable treatment of different ice types. But when I looked through the

paper, the whole method is the combination choices from different productions, which is kind of disappointed. This is NOT 'improved retracking algorithm' you wrote in the abstract.

(2) Line 68, *'found that the radar freeboard derived from the LARM has minimal errors compared'*, the authors must be very clear here that the validation from Landy et al., (2020) is based on the OIB 2011, 2012, and 2013 L4 NSIDC product.

(3) Line 80, *'Second, the calculation of radar…'*. As discussed in the previous section, it would be more appropriate to refer to it as the "radar correction coefficient" rather than the actual radar penetration. The real radar penetration is dependent on the properties of the snow and ice, not the retracking algorithm.

(4) Line 81, *'Because…, the radar freeboard errors were transferred to the radar penetration rates estimation'*, this sentence appears somewhat unaware. The empirical method is not the cause of radar freeboard errors or the existence of radar penetration rates.

(5) Line 99, '…we used LARM to replace TFMRA…', I don't see why the authors use 'replace' here since there is no consensuses on the algorithm choices until now.

(6) Line 102, 'For the snow depth, we…'. Up until this point, the authors have not highlighted any strengths of the FY3B/MWRI snow depth product. I would suggest that the authors incorporate the benefits of this product in the paragraph discussing snow depth.

(7) Line 103, *'Using the three improvements above, we ran four test cases—three individual and one combined…'*. Exercise caution when using the term "improvements" when there has been little discussion of their strengths.

(8) Line 188, *'The difference between AWI CS2 and LARM-derived radar freeboard is mainly due to the different retracking algorithms…'*, I am pretty sure this is NOT from the Landy et al., (2020), and in fact, what they did is aligning these filtering, corrections and schemes to focus on the effects from retracking algorithm itself. And they continued finding there still exist significant discrepancies from retracking itself. They NEVER said these filtering, corrections and schemes contributed to a relatively small extent. It is definitely sure that classification, waveform filtering, geophysical correction and se level tie-point interpolation exert nonnegligible effects on the final gridded radar freeboard product from each developer.

(9) Line 132, *'In this study, the MW99/AMSR2 was used in some optimization cases….'*, nstead of providing a vague explanation, the authors need to clarify where the MW99/AMSR2 dataset was used and the reasons for its inclusion. As of now, it appears that the optimization is limited to the four case studies. Therefore, calling them optimization schemes is questionable, especially considering the authors have not addressed the uncertainty associated with each product. Case studies CANNOT be equated to an optimization scheme.

(10) Line 135, I still do not understand why the authors also chose NESOSIM, SnowModel-LG, and TOPAZ4, since in Line 102, the authors mentioned the use of FY3B/MWRI. If the authors aim to compare different products to determine the best combination, they should refrain from stating that FY3B/MWRI is used for improvement in the Introduction part.

(11) Section 2.2, the whole section should have specific description of the spatial and temporal resolution used in this paper, e.g. monthly? Daily? Time span? From which month to month?

(12) Line 178, In the Data gridding section, the authors need to explain the data protocol for daily/subdaily datasets (NESOSIM, SnowModel-LG, TOPAZ4, and all observational data) and the monthly dataset (W99/AMSR2, CS-2). They should describe how these datasets are coordinated in this study, such as whether all datasets are averaged into a monthly

setting. Additionally, it is important to provide a clear explanation of the method used for spatial interpolation.

(13) Section 3.1, I have several questions about this section. As I understand it, this section calculates the radar penetration based on all observation radar/snow freeboard and CS-2 LARM radar freeboard, right? In that case, the total freeboard should be calculated from AWI IceBird and IMB ice thickness and snow depth datasets. It is necessary to specify which density is used for these calculations. Furthermore, OIB products have their own protocols for calculating total freeboard. How are these protocols coordinated fairly or placed within the same context? Additionally, since you have already used the results of MYI and FYI penetration factors based on all observations, it is unclear why these datasets are used for further validation. It does not seem fair to use them again for validation, considering they were already used for radar penetration correction.

(14) Line 258, *'The differences in radar penetration rates…'*. Once again, it should be noted that the differences in radar penetration can be explained by factors such as frequency, sensor, and period, but not solely by the spatial resolution.

(15) Line 259, *'For example, for the OIB, the radar penetration rates may be applicable only in the spring.'*, so, you did not use the OIB from October to November, right? (That's why the clear information in datasets using in the Data and Method part is very important)

(16) Line 267, 'The relationship between FYI and MYI penetration rates supports the previous studies…', It is not clear why you consider all of these relationships to be consistent. Nandan et al. (2017) deduced a depth-dependent saline snow correction factor from observations, and Landy et al. (2022) used 0.9 as a first approximation due to the difficulty of quantifying snow cover changes between May and September. It would be helpful to provide further clarification on how these studies align with your findings.

(17) Fig. 4(a). It is intriguing why the snow depth from FY3B/MWRI is higher in October compared to November. Additionally, it would be beneficial to clarify whether Figure 4 represents Arctic basin-scale mean values. If so, it is puzzling why radar freeboard and thickness are larger in October than in November. Providing possible explanations for these observations would be valuable.

(18) From the Table 2 and Section 3.3, the improvements observed among different cases are only reflected in the RMSE, which is expected since you corrected or generally reduced the values based on the observations. However, it is frustrating that these four cases differ in at least two products, making it challenging for readers to make direct comparisons.

(19) Line 313-314, I assume you consider AWI CS2 as your baseline and aim to determine whether the results are better than AWI CS2. If that is the case, you should provide this context from the beginning. However, I have some concerns since the work now uses a completely different algorithm and observed-corrected coefficient for comparison, which may be unfair to AWI CS2.

(20) Line 310-333. Among the in-situ observations, only CryoVex provides actual independent validation. Upon closer examination of the third column in Figure 6, all cases show high correlation coefficients, and the combination cases reduce the RMSE by over 23% compared to AWI CS2. Therefore, there does not seem to be a significant improvement in the LARM+FY3B/MWRI+RP choice compared to the other cases. The differences lie in the slopes, but it is unclear whether you placed the retrieved data on the x-axis and the in-situ/real data on the y-axis. Mathematically, the x-axis in linear fitting should represent the true/validation data, or else there might be considerable uncertainty in data validation. Therefore, if you were to switch the axes, the slope would likely be different. Additionally, there is a concern that the LARM+FY3B/MWRI+RP combination might result in significant underestimation of sea ice thickness.

(21) Section 4. It is unclear what the main takeaway is from the entire Section 4, where numerous pictures and discussions focus on the differences between each combination, ranging from spatial patterns to spatial-temporal trends. Since the previous parts have already discussed the improvement in the optimization case, it seems unnecessary to include all combinations here and analyze their differences. This approach might cause readers to lose focus and miss the main points. Additionally, in the abstract, Section 4 is summarized in just one sentence stating that MYI ice thickness is decreasing, which is already quite obvious since lowering the radar correction would naturally reduce the ice thickness. To simplify the paper, it might be better to move Figures 9 and 10 to the Supplementary section.

(22) Line 455-459, by combining Equations (8) and (9), it is evident that there is a linear relationship between density and radar penetration, with ice density having a larger coefficient than snow density. It would be beneficial to see more uncertainty quantification, such as considering the combined effects of IMB and LARM, and the uncertainties associated with radar penetration derived from observed ice thickness, snow depth, ice density, snow density, and radar freeboard.

(23) Line 474-477. The temporal sampling is a significant concern in the paper. As mentioned, only IMB data was used from October to February, which raises questions about representativeness and could compromise the results. It would be helpful to provide further explanation on this issue.

(24) Figure 15, I am very curious how the radar penetration rates vary from year to year. Including such information in the figure would be valuable.

(25) Section 5.2. Like I suggested before, it is important to combine the sensitivities from all parameters. However, it is unclear whether this section focuses on the sensitivity of radar penetration or sea ice thickness. If it is about radar sensitivity, then it is unnecessary to bring up other snow products and their effects, as you have already compared them earlier. It would be better to concentrate on the uncertainties of the FY3B/MWRI snow product in relation to radar penetration. If you also want to discuss the sensitivity study of ice thickness, you should systematically address the uncertainties associated with LARM radar freeboard, FY3B/MWRI snow product, density choice, and derived radar penetration. Additionally, in Figure 16, you introduce another validation on sea ice thickness, which is confusing. It is unclear whether this figure is part of the sensitivity analysis or a validation study.

(26) Section 5.3, once again, it is crucial to clearly distinguish between sensitivity studies and validation studies. When discussing the uncertainty of density on sea ice thickness results, it is important to recognize that this pertains to the density choice and its impact on ice thickness. You have already validated the results above and concluded that LARM+FY3B/MWRI+RP is the optimization case. Therefore, please utilize the validated results from earlier and avoid reintroducing these combinations here. Otherwise, it will confuse readers and undermine the confidence and trustworthiness of the previous results. Moreover, it is not appropriate to select densities or refer to them as an "updated density scheme" solely based on having lower RMSE than others after several rounds of validation. We want the paper to avoid cherry-picking results.

---

## Author Comment (AC2)

**Responses to Reviewer**

**Authors' response:** We appreciate Reviewer 1 for her/his dedicated comments. We have made significant revisions to the content and structure of the original manuscript to ensure it meets the standards of The Cryosphere. The original referee comment is in black, and our replies are written in blue.

**Major comment 1.** This work seeks to address several open research questions surrounding the amount of snow on sea ice, the ability of the CryoSat-2's radar waves to penetrate snow, and radar waveform interpretation. Reducing one of these uncertainties may move the subsequently retrieved value further away from "the truth" due to the presence of compensating biases, so it makes sense to consider several sources of uncertainty at once and then optimise. This work is therefore well motivated.

In this submission the authors have interchanged several aspects of the radar-freeboard to sea-ice-thickness processing chain. In particular, they have used two different "off the shelf" radar freeboard products (AWI and Bristol/UIT), several snow products (AMSR/W99, SnowModel-LG, NESOSIM, TOPAZ4, FY3B/MWRI), and have compared the assumption of full radar penetration of the assumed snowpack to an ice-type dependent radar penetration factor. These radar penetration factors were derived from comparison to several separate (but often not independent) data sources such as airborne estimates of sea ice freeboard/thickness and drifting ice-mass-balance buoys (Fig. 3). Having made these various interchanges, the authors appear to compare their new SIT data to similar evaluation data as before, which seems like a conflict between training and testing (Table 2; Fig. 5)

**Authors' response:** The reviewer mentioned that we used some kinds of dataset to calculate the radar penetration factors and used the same datasets to evaluate the derived sea ice thickness. We agree with the reviewer's comment. In the revised manuscript, the airborne and buoy measurements are no longer used to calculate radar penetration rates. We used the total freeboard from ICESat-2 (IS2), snow depth from FY-3B, the radar freeboard from LARM, and snow densities from SnowModel-LG driven by ERA5 to recalculate radar penetration rates. The new radar penetration rates are more representative which cover the most regions of Arctic Ocean. Therefore, conflicts between the training and test datasets can effectively be avoided. The details are as follows:

The radar penetration model proposed in this study is reserved.

$$\alpha = \frac{c_s\,(h_f - h_{fr})}{c \times h_s},\tag{1}$$

Where $h_f$ and $h_s$ are the total freeboard and snow depth, and $h_{fr}$ is the radar freeboard. $c$ is the speed of light ($3 \times 10^8$ m s$^{-1}$), and $c_s$ is the radar propagation speed in the snow. In this study, $c_s$ was obtained from a snow density ($\rho_s$)-dependent parameterization: $c_s = c(1 + 0.51\rho_s)^{-1.5}$ m s$^{-1}$ (Ulaby et al., 1982).

In addition, the reviewer suggested us to discuss our optimised radar penetration factors with reference to Nab et al. (2023). In the Eq (1), the daily optimal interpolation CryoSat-2 (CS2) radar

freeboard (LARM) and IS2 total freeboard are provided by Nab et al. (2023). They derived pan-Arctic radar penetration rates based on the relationship between radar freeboard (CPOM and LARM) and SnowModel-LG snow depth. They successfully demonstrated the winter Ku-band radar scattering above the snow-ice interface, giving us confidence to promote this work (in this revised manuscript). Similarly, regarding Eq. (1), once the snow depth and total freeboard are determined, estimating the radar penetration rate at the basin scale becomes possible.

The reasons for selecting these datasets are as follows:

- *CS2 radar freeboard (LARM):* Landy et al. (2020) developed a Lognormal Altimeter Retracker Model (LARM) and found that the radar freeboard derived from the LARM has lower errors compared with other retrackers. It should be acknowledged that there are still some difficulties in interpreting radar waveforms (Ricker et al., 2014), which can introduce potential bias in LARM-derived radar freeboard and further propagate to radar penetration rate calculation (Nab et al. 2023 also faces similar challenges).

- *IS2 total freeboard:* The NASA's ICESat-2 mission was launched in September 2018 with the primary goal of monitoring the height of ice sheets and sea ice with high accuracy, leading us to better understand changes in polar regions (Markus et al., 2017). ICESat-2 is capable of measuring the vertical distance from snow-covered surface on sea ice to the sea surface, which is total freeboard (or snow freeboard), and enables the estimation of sea ice thickness by assuming hydrostatic equilibrium (Petty et al., 2020). The total freeboard reached an uncertainty of 2–4 cm when compared with NASA's Operation IceBridge (OIB) airborne measurements data (Kwok et al., 2019).

- *FY3B/MWRI snow depth:* The FY-3B meteorological satellite is a second-generation polar-orbiting meteorological satellite from China launched in November 2010. The FY3B/MWRI snow depth was developed by Li et al. (2021) with a spatial resolution of 12.5 km × 12.5 km and is available for 2013–2020, encompassing the entire sea ice growth season. This new snow depth data show smaller biases with the OIB-derived snow depth, with a mean difference of 2.89 cm on FYI and 1.44 cm on MYI.

- *SnowModel-LG snow density*: SnowModel-LG is a Lagrangian snow-evolution model developed to simulate snow depth and density on a pan-Arctic scale (Liston et al., 2020). The model mainly uses two typical atmospheric reanalysis data (i.e., ERA5 and MERRA2), and simulates full surface, internal energy, and mass balances within a multilayer snowpack evolution system. Stroeve et al. (2020) conducted an assessment analysis of SnowModel-LG and showed that it can well capture the spatial and seasonal variability of Arctic snow depth and snow density.

   The new monthly radar penetration rates at the pan-Arctic scale are shown in Figs. 1-2. Each effective penetration rate was derived by averaging the monthly penetration rates from 2018 to 2020. In particular, non-physical data points were excluded from the calculation, i.e., radar penetration rates above 1 or below 0. Generally, radar penetration rates increase from fall to spring. Note that radar penetration rates tend to be lower in some marginal seas when

compared to the central Arctic region.

[Figure]

Fig. 1. Monthly mean radar penetration rates. (a) spatial distribution and (b) probability density characteristics

[Figure]

Fig. 2. Regional mean of radar penetration rates. (a) describes the sub-regions of the Arctic Ocean, including the Central Arctic (CA), East Siberian Sea (ESS), Laptev Sea (LS), Kara Sea (KS), Barents Sea (BS), East Greenland (EG), Baffin Bay (BB), Beaufort Sea (BS) and Chukchi Sea (CS). (b) monthly mean radar penetration for each subregion.

**Major comment 2.** To be honest, I found it difficult to identify any clear conceptual or practical advances in this work. I was also left somewhat doubting whether the claims in the abstract and conclusions were supported by the results. It seems to me that rather than a "comprehensive optimisation", this work represents several experimental recombinations of different existing components in the processing chain. Furthermore, the authors' recombinations are potentially improving one metric of skill (e.g. RMSE) at the expense of another metric (e.g. correlation R value). I also note that not all combination possibilities were explored.

**Authors' response:** In the revised manuscript, the title has been modified to: "Assessment of radar freeboard, radar penetration rate, and snow depth for potential improvements in CryoSat-2 sea ice thickness retrieval". In this study, we focus on the impacts of radar freeboard, radar penetration rate, and snow depth on retrieving CryoSat-2 sea ice thickness and investigate the potential improvements in sea ice thickness. We don't think our experiments improved one metric of skill (e.g. RMSE) at the expense of another metric (e.g. correlation R value). Based on the new assessment results of sea ice thickness compared with OIB L4 (Fig. 3) and CryoVEX-EM (Fig. 4), the correlation R value can be kept with the similar results with the original AWI CS2, even in some cases, the R values are higher. In addition, the reviewer mentioned that not all combination possibilities were explored. As described before, our purpose is to explore the impacts of radar freeboard, radar penetration rate, and snow depth on retrieving CryoSat-2 sea ice thickness and investigate the potential improvements in sea ice thickness. These three factors are independent. The three individual cases by focusing on the retracking algorithm, radar penetration rate, and snow depth, and one combined case are sufficient to confirm the potential impacts of these factors on the improvements in sea ice thickness. We aim to provide some feasible schemes to the optimizations of sea ice thickness derived from AWI CS2.

[Figure]

Fig 3. Validation of sea ice thickness improvement with the OIB L4. The correlation coefficient (R), root mean square error (RMSE), and the number of samples (N) are shown in each subfigure. The solid black line indicates the best fitting line, and the solid red line indicates the scatter fitting line (the fitting equation is also shown in each subfigure).

[Figure]

Fig 4. Validation of sea ice thickness improvement with the CryoVEX-EM. The correlation coefficient (R), root mean square error (RMSE), and the number of samples (N) are shown in each subfigure. The solid black line indicates the best fitting line, and the solid red line indicates the scatter fitting line (the fitting equation is also shown in each subfigure).

**Major comment 3.** I was also concerned about the treatment of OIB QuickLook data as "in situ validation" (Table 2; Figure 5), since that particular data set has several known issues and is of course an airborne remote sensing product (not in-situ). Similarly, AWI IceBird and CryoVEX airborne data are also referred to and treated as "in situ validation", but are of course also the results of airborne remote sensing with their own biases and uncertainties. Particular to CryoVEX, it is not appropriate to use Ku-band ASIRAS data processed with the assumption of full radar penetration to then work out CS2 penetration factors

**Authors' response:** We agree with the reviewer's suggestion. In the revised manuscript, we no longer describe the airborne measurements as in situ observations. Instead, we describe them as airborne-derived products. In the original manuscript, we did not use CryoVEX data to derive the radar penetration rate; we only used the total thickness measured by CryoVEX-EM to evaluate the sea ice thickness data.

**Major comment 4.** Relatedly, the issue of how representative an ice-mass-balance buoy's measurement is of the ice sampled by (a) a CryoSat-2 SAR footprint, or (b) the ice in a 25×25 km grid cell such as those analysed here, was mentioned (Sect. 2.4) but not meaningfully tackled. These issues strongly affect the suitability of IMBs for radar-altimetry evaluation. More broadly, the extent to which the authors may be optimising towards an uncertain or even biased truth is not discussed.

**Authors' response:** The notable difference in spatial resolution between buoys and satellite

observations remains a big challenge in contemporary research. Recently, Koo et al. (2021) matched buoy data with modal values of sea ice thickness distribution derived from satellite tracks within a certain radius and found good agreement. However, we cannot access the LARM-derived along-track radar freeboard data to test this approach. Therefore, in the revised manuscript, we no longer use the buoy data for assessing sea ice thickness data.

**Major comment 5.** Finally, it was not possible for me to reproduce this analysis since it relies substantially on an unpublished snow dataset from the FY3B radar altimeter (which one coauthor has previously led a publication on). Since the manuscript relies so heavily on publicly available data, I thought it was a shame not to act in this spirit. The reason for the data not being made available was not given (despite an explanation being required by the Copernicus data policy). On a related note, no code was made available with this manuscript for review. These factors will limit the impact of any publication, and also limit my confidence as a reviewer.

**Authors' response:** I am afraid the reviewer has some misunderstandings regarding the data and code availability. In the email communications with the developers from AIW CS2, we have expressed our willingness to share all of the data and codes with the community. The reason why the manuscript did not show the publicly available data and code is that this is an unpublished manuscript. Before accepted, there must be some revisions to do. Therefore, our codes and data probably have to be changed. We prefer to share a final version of codes and data after the manuscript is accepted. In addition, the FY3B/MWRI snow depth data can be accessed at http://coas.ouc.edu.cn/pogoc/2021/0609/c9718a335873/page.htm.

**Major comment 6.** It is therefore my recommendation that this manuscript should be rejected, and a similar analysis reperformed and resubmitted in service of a more carefully constructed research question. I appreciate this might be received as an overly negative recommendation, but I believe our community has a long way to go until we can meaningfully conduct the comprehensive optimisation marketed in this manuscript. Such an exercise would require reprocessing of several key datasets by many different research groups for a truly "apples to apples" comparison, not to mention a deep new look at the sampling biases, systematic uncertainties and independence of airborne and in-situ evaluation data.

**Authors' response:** We sincerely appreciate the reviewer's constructive comments. Those comments are all valuable and very helpful for revising and improving our paper. In the revised manuscript, we have followed the reviewer's suggestion to separate the datasets into two parts: one for the calculation of radar penetration rates, and the other for the evaluations of sea ice thickness. In the calculation of radar penetration rates, we no longer used the previous datasets, but used the total freeboard from ICESat-2 (IS2), snow depth from FY-3B, the radar freeboard from LARM, and snow densities from SnowModel-LG driven by ERA5 to replace them. In the evaluations of sea ice thickness, we don't use the data of OIB Quick Look, IMB, and AWI IceBird as the reviewer suggested. We used the data of OIB L4 and CryoVEX-EM. Hopefully the revision could meet the reviewer's requirement.

Major Concerns

Operation IceBridge snow depth, freeboard and thickness data are not "in-situ validation"

Firstly (and this is a somewhat trivial point), they are not "in-situ" because they are taken from an aircraft. They are much better described as data products from airborne radar (and sometimes lidar) remote sensing.

**Authors' response:** Same comment as before. In the revised manuscript, we revised it to "airborne-derived product".

Less trivially, because they're radar-based products, they're subject to significant uncertainties involving radar penetration of snow, sea ice roughness, and particularly sidelobes. Much has been written about OIB snow depth retrievals, and the quicklook product that the authors have relied upon in several sections is notoriously poor. In particular, it has relatively poor performance over multiyear ice (King et al., 2015), and the underlying algorithm suffers from a persistent issue of misidentifying range sidelobes from the snow-ice interface as the snow-air interface (Kwok and Maksym, 2014; Kwok and Haas, 2015). Figure 4 of Kwok et al. (2017) illustrates that the GSFC-NK product (which I think corresponds to the OIB QL product?) shows persistently low snow depths. In any case, the figure definitely shows that several very different interpretations of the OIB raw data exist, and therefore any individual snow or ice product cannot represent some kind of "in-situ" truth.

Even in the case where there are no snow-penetration, roughness or sidelobe-related biases, OIB sea ice thickness estimates still rely on exactly the same uncertain conversion of freeboard to thickness as CryoSat2, so cannot form an independent benchmark for optimisation. For instance, I was hoping for some analysis of how the snow and ice densities used the in OIB hydrostatic conversions compared to those being used in the satellite conversions at hand, but that was glossed over. This type of material must be foundational to any activity that aims to "optimise" satellite retrievals to these OIB products. Otherwise the optimised product be biased in all the ways in which the OIB products are biased. Along these lines, describing OIB thickness products as "observations" is also dubious. The confidence with which you can "observe" SIT even with accurate measurements of snow depth and freeboard is nicely documented by Alexandrov et al. (2010; see numbers in second half of abstract).

**Authors' response:** We have followed the reviewer's suggestion and don't use OIB Quick Look as the evaluation data. We use OIB L4 to replace it.

CryoVEX Data

The authors need to be much more specific and thoughtful about how CryoVEX data on total thickness are generated – it's not enough to say "airborne electromagnetic sensors". If the CryoVEX method is similar to CryoSat-2, it will suffer from similar biases as CryoSat-2 and therefore is unsuitable for independent evaluation. Furthermore, it appears that the authors have used a very limited subset of CryoVEX data from 2014. Aside from limiting the applicability of their results, my understanding is that SIT for the CryoVEX'14 campaign is from combination of the Ku-band ASIRAS system and the ALS system (Hvidegaard et al., 2015). Given the study by Willatt et al. (2011) on ASIRAS radar penetration through snow on sea ice, I think a lot more consideration needs to be given to the positioning of this data as some kind of objective truth that can be used to optimise satellite data. It's also worth pointing out that if you look at Fig. 9 from

Garnier et al. (2021) which analyses CryoVEX 2017 data, it's clear that the difference between the ALS and ASIRAS data is often negative towards the end of the track, further indicating that the "off-the-shelf" approach to CryoVEX ALS/ASIRAS data in this manuscript may need considerable further scrutiny. At an absolute minimum, the authors should understand they cannot get at the CS2 radar penetration factor using Ku-band CryoVEX data that assumes a 100% penetration factor from ASIRAS. When comparing CS2 freeboards to ASIRAS freeboards derived from the assumption of a 100% RP, is it any wonder that we find high CS2 radar penetration factors?

**Authors' response:** I am afraid the reviewer has some misunderstandings here. Firstly, the CryoVEX data we utilized in our study is the total thickness measurements obtained through airborne electromagnetic detection in 2014, not ALS/ASIRAS data as the reviewer described. Therefore, CryoVEX data was not used to calculate radar penetration rate in our study.

Ice mass balance buoys

There are major issues with using single IMBs to validate measurements by radar altimeters such as CryoSat-2. IMBs are generally deployed on level ice, and are of course point measurements. This is in contrast to as CryoSat-2 footprint which is several hundred metres long and several kilometers wide depending on the ice roughness. And of course this is very different to the 25×25km scale on which this manuscript's analysis is conducted. It's also worth considering that IMBs essentially break as soon as the ice on which they're situated begins to grow dynamically, and dynamic growth can be quite a significant contributor to the thickness in a CS2 footprint and also in a 25 km grid cell. So much more serious consideration needs to be given to how and whether the IMB data correspond to that collected by CryoSat-2, and whether they are suitable for a tuning/optimisation exercise such as this.

**Authors' response:** As mentioned above, the IMB data is no longer used in the revised manuscript.

Radar Penetration Factors

I assume 0.77, 0.96, and 0.91 for the penetration factors were generated from the means of the histograms in Fig. 3A (not the modes, and not the mean of the Gaussian fits) – this should be clarified. More information is also required on how the "all data" histograms are constructed. This is because there are presumably many more data points from some sources than from others: were their contributions weighted for measurement reliability or quantity? It seems like the only reason that FYI penetration is significantly lower than MYI is because of the contribution of IMB data (panel f), but this is probably related to the fact that IMBs have a distinct sampling bias relative to both the other data and their surrounding environment (discussed above). So is it reasonable to allow the derived penetration factor to be so strongly influenced by the IMB contribution? This again goes back to how different sources of evaluation data should be weighed against each other in terms of reliability and quantity.

How do the authors justify the width of the uncertainty bars in Figure 15? The shaded bars around the line seem surprisingly thin for a quantity that's eluded the CryoSat community so effectively over the last decade. Could it really be the case that the penetration-factor consistently goes up in January and then back down in Feb by a magnitude considerably larger than the uncertainty bars? What is the spatial range of applicability of these statistics? How are the printed numbers of distinct in-situ observations generated? After all, there are hundreds of thousands of SnowRadar/ASIRAS

waveforms and ALS spot heights going into this analysis, So I guess the figures in the thousands correspond to the number of contributing 25 km×25 km grid cells? If more observations go into an aggregated 25x25 km data point, then we should probably weight that data point's reliability as being higher than a grid cell that contains only a few data points. Has this been factored into the uncertainty analysis?

**Authors' response:** As mentioned above, we have changed the datasets to calculate the radar penetration rates.

I have several further questions about the radar penetration factor analysis, both concerning how it was done and how it is reported, but will leave it here.

Radar freeboard data and retracking

I don't think it's legitimate to claim to have applied a "comprehensive optimisation of an improved retracking algorithm" (from the abstract) when no retracking takes place in the manuscript. This is misleading to the casual reader. A comparison has not been made between retracking algorithms (TFMRA & LARM), but instead between two radar freeboard products (AWI & Bristol/UIT). To imply that differences in the RF products only represents differences in the retracking approach is a risky business. The authors state "although the different classifying waveforms, geophysical corrections, and sea level tie-point interpolation also contribute to a relatively small extent (Landy et al., 2020)." That's not exactly my understanding from reading that paper; Figure S3 shows there definitely are some differences that derive from the "in-house" treatments of AWI vs Bristol/UIT. That's the reason that Landy et al. 2020 emulated the other retrackers with the same processing: to learn about the retrackers in isolation. I think that's the approach that should be taken here if the retrackers are to truly be "optimised" rather than the radar freeboard products

**Authors' response:** We agree with the reviewer's suggestion. In the revised manuscript, we don't describe the "improved retracking algorithms", and used "LARM radar freeboard product".

Optimisation Metrics

Firstly, the metric referred to as R needs some clarification. Is this the Pearson product-moment correlation coefficient? R is also often used for the coefficient of determination, which essentially captures the data's deviation from the line y=x (i.e. it also captures the bias, where Pearson does not). Using the product-moment correlation coefficient has its pitfalls. i.e. with heavy optimisation for Pearson you end up capturing the variance of your evaluation data, but at the cost of increasing your bias. This issue should be handled explicitly. These concepts are at the core of the "bias-variance tradeoff", and allied concepts in optimisation such as overfitting. While RMSE is sensitive to the bias, it also can be large when spread about the y=x line is large and the bias is small. So I think the metrics against which the processing chain is optimised need much more careful consideration when defining the optimisation exercise

**Authors' response:** R used in the manuscript is the Pearson correlation coefficient. This is the common correlation coefficient in most studies. As described above, the correlation R value can be kept with the similar results with the original AWI CS2, even in some cases, the R values are higher. In addition, in the revised manuscript, we added the mean error (ME) and mean absolute error (MAE) to evaluate the improvement of sea ice thickness.

Some More Minor Things

The effect of reducing the radar penetration factor from unity to some fractional value is basically to reduce the subsequently calculated ice freeboard and thus the derived thickness. This does the same thing as assuming a deeper snowpack. Since the authors are varying the snow and the RP at the same time, they should grapple with this conflict explicitly: when we observe the sea ice to be "too thick" relative to our evaluation data, we often don't know whether it's because our assumed RP is too high, or whether our assumed snow is too thin. How can we optimise in light of this? One option is to just assume that airborne surveys of snow depth are correct, and use that snow depth to then optimise the RP. The authors haven't done this, but it might be interesting? But it does take a bit of a leap of faith regarding airborne snow depths.

**Authors' response:** In the individual case with corrected radar penetration rate (TFMRA + MW99/AMSR2 + RP), the radar penetration rate is calculated from airborne data, assuming that total freeboard and snow depth derived from airborne measurements somewhat represent the "true value". This is consistent with the reviewer's description.

Title: I think we should steer clear of subjective adjectives like "comprehensive" in titles. One could easily argue that this analysis is not comprehensive as it uses a fairly limited set of in-situ sea ice thickness measurements relative to those available, and uses limited subsets of CryoVEX and IceBird data. I recommend removal of this word

**Authors' response:** The original manuscript's title has been modified to: "Assessment of radar freeboard, radar penetration rate, and snow depth for potential improvements in CryoSat-2 sea ice thickness retrieval".

L42: "Recent advances in satellite altimetry began in 2003." – This is a very subjective sentence and I'm not sure what it adds. Envisat began operating in 2002 for instance. What defines "Recent" here?

**Authors' response:** We have deleted this sentence.

L68: Lower errors, not minimal.

**Authors' response:** We have revised it.

L140: I think it would be better to cite the official NSIDC source for these data: https://nsidc.org/data/nsidc0758/versions/1

**Authors' response:** Revised.

L509: Perhaps I've missed this, but I don't think the authors have specified whether they're using the ERA5 or MERRA2 run of SnowModel? When comparing SnowModel and NESOSIM it's of course important to make sure they're forced by similar data, or else you're basically just comparing reanalysis precipitation data rather than the models themselves. Same with TOPAZ – I assume this is driven by some ERA-type product?

**Authors' response:** The SnowModel-LG data used in this study is driven by ERA5 and TOPAZ4 data is driven by ERA- interim.

Fig. 1: There's no reference to the bathymetry in this paper, so I think shading it into this plot confuses things.

**Authors' response:** Revised.

Fig. 2: If you label the "Reference ellipsoid" in this figure you should reference/explain it in the text.
**Authors' response:** We have removed the "Reference ellipsoid" from Fig. 2.

L379: Think this sentence needs rewording for clarity.
**Authors' response:** Revised.

L447: The authors should discuss their optimised radar penetration factors with reference to Nab et al. (2023), who derived RP factors based on the relationship between radar freeboard and SnowModel-LG depths.
**Authors' response:** Added.

L594: I was disappointed to see that the snow depth data from Li et al. (2021) are not made publicly available, so many of the results presented in this paper are not replicable by either me as a reviewer, or the community at large. As well as limiting my confidence as a reviewer, this will limit the impact of the paper since the supposedly optimised products are not available or reproducible in future. The Copernicus Publications data policy states "if data are not publicly accessible, a detailed explanation of why this is the case is required", and I can't see why this was ignored. Furthermore, no code was made available with the manuscript, which also limits my confidence in recommending the paper for publication.
**Authors' response:** Same comment as before. Please check the answer in Major comment 5.

Reference

Armitage, T. W. and Ridout, A. L.: Arctic sea ice freeboard from AltiKa and comparison with CryoSat-2 and Operation IceBridge, Geophysical Research Letters, 42, 6724-6731, 2015.

Koo, Y., Lei, R., Cheng, Y., Cheng, B., Xie, H., Hoppmann, M., Kurtz, N. T., Ackley, S. F., and Mestas-Nuñez, A. M.: Estimation of thermodynamic and dynamic contributions to sea ice growth in the Central Arctic using ICESat-2 and MOSAiC SIMBA buoy data, Remote Sensing of Environment, 267, 112730, 2021.

Kwok, R., Kacimi, S., Markus, T., Kurtz, N., Studinger, M., Sonntag, J., Manizade, S., Boisvert, L., and Harbeck, J.: ICESat-2 surface height and sea ice freeboard assessed with ATM lidar acquisitions from Operation IceBridge, Geophysical Research Letters, 46, 11228-11236, 2019.

Landy, J. C., Petty, A. A., Tsamados, M., and Stroeve, J. C.: Sea Ice Roughness Overlooked as a Key Source of Uncertainty in CryoSat-2 Ice Freeboard Retrievals, Journal of Geophysical Research-Oceans, 125, 2020.

Li, L., Chen, H., and Guan, L.: Retrieval of Snow Depth on Arctic Sea Ice from the FY3B/MWRI, Remote Sensing, 13, 2021.

Liston, G. E., Itkin, P., Stroeve, J., Tschudi, M., Stewart, J. S., Pedersen, S. H., Reinking, A. K., and Elder, K.: A Lagrangian Snow-Evolution System for Sea-Ice Applications (SnowModel-LG): Part I-Model Description, Journal of Geophysical Research-Oceans, 125, 2020.

Markus, T., Neumann, T., Martino, A., Abdalati, W., Brunt, K., Csatho, B., Farrell, S., Fricker, H.,

Gardner, A., and Harding, D.: The Ice, Cloud, and land Elevation Satellite-2 (ICESat-2): science requirements, concept, and implementation, Remote sensing of environment, 190, 260-273, 2017.

Nab, C., Mallett, R., Gregory, W., Landy, J., Lawrence, I., Willatt, R., Stroeve, J., and Tsamados, M.: Synoptic variability in satellite altimeter-derived radar freeboard of Arctic sea ice, Geophysical Research Letters, 2023. e2022GL100696, 2023.

Petty, A. A., Kurtz, N. T., Kwok, R., Markus, T., and Neumann, T. A.: Winter Arctic sea ice thickness from ICESat-2 freeboards, Journal of Geophysical Research: Oceans, 125, e2019JC015764, 2020.

Ricker, R., Hendricks, S., Helm, V., Skourup, H., and Davidson, M.: Sensitivity of CryoSat-2 Arctic sea-ice freeboard and thickness on radar-waveform interpretation, The Cryosphere, 8, 1607-1622, 2014.

Stroeve, J., Liston, G. E., Buzzard, S., Zhou, L., Mallett, R., Barrett, A., Tschudi, M., Tsamados, M., Itkin, P., and Stewart, J. S.: A Lagrangian snow evolution system for sea ice applications (SnowModel-LG): Part II—Analyses, Journal of Geophysical Research: Oceans, 125, e2019JC015900, 2020.

Ulaby, F., Moore, R., and Fung, A.: Microwave remote sensing: Active and passive. Volume 2-Radar remote sensing and surface scattering and emission theory. 1982.

---

## Author Comment (AC3)

**Responses to Reviewer**

**Authors' response:** We appreciate Reviewer 2 for her/his dedicated comments. We have made significant revisions to the content and structure of the original manuscript to ensure it meets the standards of The Cryosphere. The original referee comment is in black, and our replies are written in blue.

**Major comment.** The paper aims to address several difficulties encountered in CS-2 (CryoSat-2) data analysis, including waveform retracking, snow depth estimation, and radar penetration. It is noteworthy that the paper summarizes a substantial amount of observational data for validation, compares snow reconstructions, and employs two CS-2 retracking algorithms to generate an ice thickness dataset. However, I share Reviewer 1's opinion that the paper lacks comprehensive optimization. Instead, it leans towards combining algorithms and products in a somewhat selective manner, which is inappropriate for a scientific paper, particularly one focused on datasets or products. Additionally, there is a consistent mixing of sensitivity studies and validation studies, making the paper challenging to read and diminishing its overall credibility. Furthermore, there is a critical need for clarification and correction regarding the definition of radar penetration and potential misunderstandings arising from previous papers.

I acknowledge the usefulness and instructive nature of referring to Armitage et al. (2015) to gain a quick understanding of the priorities concerning the CS-2 radar penetration problem. However, the algorithms employed in the current paper are essentially the same as those presented in Armitage et al. (2015), which calls for innovation and reorganization given the passage of several years. It is important to exercise caution, particularly in two areas: (1) the potential misuse and conflict arising from using training data for radar penetration estimation and validation data for sea ice thickness validation, an issue that Reviewer 1 has already highlighted; and (2) a genuine misunderstanding of how the paper defines the radar penetration factor. The authors seem to suggest that the radar penetration factor depends on the waveform retracking algorithms and snow depth product.

However, this may lead to significant misunderstandings since the radar penetration effects are a result of snow and ice scattering, which, in turn, depend on the properties of the upper snow and ice layers, such as wetness, density, and grain size. Therefore, the radar penetration of CS-2 should be based on the properties of the snow and ice itself, rather than solely relying on the waveform algorithm or snow product. If the authors aim to address the radar freeboard penetration or snow scattering problem, a more appropriate and physically robust approach would be to follow the methodology outlined by Slater et al. (2019), where the penetration depth over Greenland was derived, or refer to the study by Kurtz et al. (2014), which used a model to address the uncertainty of radar penetration. If the authors genuinely seek to reduce bias in a specific radar freeboard estimation, I would suggest the following steps: (1) select an appropriate snow product; (2) choose an appropriate radar freeboard estimation (or retracking algorithm); and (3) based on current in-situ observations, correct the selected radar freeboard bias. In this approach, the fundamental tasks involve selecting the proper snow product and radar freeboard estimation, which have not been

adequately addressed in the paper. Additionally, the datasets used for correction and validation purposes lack clarity. Therefore, I strongly recommend that the authors consider changing the term "radar penetration factor" to "radar correction coefficient" to better align with their protocols and enhance the overall structure of the paper.

**Authors' response:**
(1) In the revised manuscript, the title has been modified to: "**Assessment of radar freeboard, radar penetration rate, and snow depth for potential improvements in CryoSat-2 sea ice thickness retrieval**". In this study, we focus on the impacts of radar freeboard, radar penetration rate, and snow depth on retrieving CryoSat-2 sea ice thickness and investigate the potential improvements in sea ice thickness.

(2) The essence of the AR15 method, which is used to calculate radar penetration rate, is derived from the radar freeboard correction equation (Eq.1-3), making it an indirect method. In other words, this is not derived from the physical mechanism (e.g., properties of snow and ice). We have added clarifications to the revised version.

$Ice\ Freeboard\ =\ Radar\ Freeboard\ +\ Speed\ Correction\ +\ Penetration\ Correction$

$$h_{\mathrm{fi}}\ =\ h_{\mathrm{fr}}\ +\ h_{\mathrm{c}}\ +\ h_{\mathrm{p}}, \tag{1}$$

$$h_{\mathrm{fi}}\ =\ h_{\mathrm{fr}}\ +\ \left(\frac{c}{c_{\mathrm{s}}}-1\right)h_{\mathrm{p}}\ +\ h_{\mathrm{p}}, \tag{2}$$

$$h_{\mathrm{fi}} = h_{\mathrm{fr}} + \left(\frac{c}{c_{\mathrm{s}}}-1\right)\alpha h_{\mathrm{s}} + (\alpha-1)h_{\mathrm{s}}. \tag{3}$$

Where $h_{\mathrm{fi}}$ and $h_{\mathrm{fr}}$ are the sea ice freeboard and snow depth, $h_{\mathrm{c}}$ and $h_{\mathrm{p}}$ are the radar speed and penetration correction terms. $c$ is the speed of light ($3 \times 10^8$ m s$^{-1}$), and $c_{\mathrm{s}}$ is the radar propagation speed in the snow. $\alpha$ is the radar penetration rate, which can be further expressed as

$$\alpha = \frac{c_{\mathrm{s}}\,(h_{\mathrm{f}}-h_{\mathrm{fr}})}{c \times h_{\mathrm{s}}}. \tag{4}$$

When the parameters in Eq.4 are more accurate, a more "realistic" radar penetration rate is obtained. **Following the reviewers' comments, we have uniformly revised the radar penetration rate to radar correction rate.**

(3) The reviewers mentioned that we used some kinds of dataset to calculate the radar penetration factors and used the same datasets to evaluate the derived sea ice thickness. **We agree with the reviewer's comment. In the revised manuscript, the airborne and buoy measurements are no longer used to calculate radar penetration rates.** We used the total freeboard from ICESat-2 (IS2), snow depth from FY-3B, the radar freeboard from LARM, and snow densities from SnowModel-LG driven by ERA5 to recalculate radar penetration rates (Fig.1-2). The reasons for choosing these parameters are given in the response to reviewer 1.

[Figure]

Fig. 1. Monthly mean radar penetration rates. (a) spatial distribution and (b) probability density characteristics

[Figure]

Fig. 2. Regional mean of radar penetration rates. (a) describes the sub-regions of the Arctic Ocean, including the Central Arctic (CA), East Siberian Sea (ESS), Laptev Sea (LS), Kara Sea (KS), Barents Sea (BS), East Greenland (EG), Baffin Bay (BB), Beaufort Sea (BS) and Chukchi Sea (CS). (b) monthly mean radar penetration for each subregion.

Based on the aforementioned concerns and the following comments, I suggest rejecting this manuscript. Here are my main comments:

(1) Line 15, 'applying a comprehensive optimization of an improved retracking algorithm, corrected radar penetration rate'. When reading this line, it gives the impression that the paper will introduce a genuinely improved retracking algorithm, such as better waveform fitting or more reasonable treatment of different ice types. But when I looked through the paper, the whole method is the combination choices from different productions, which is kind of disappointed. This is NOT 'improved retracking algorithm' you wrote in the abstract.

We agree with the reviewer's suggestion. In the revised manuscript, we corrected the major goal of the paper, which was to assess the effects of radar freeboard, radar penetration rate, and snow depth on the estimation of CryoSat-2 sea ice thickness.

(2) Line 68, 'found that the radar freeboard derived from the LARM has minimal errors compared', the authors must be very clear here that the validation from Landy et al., (2020) is based on the OIB 2011, 2012, and 2013 L4 NSIDC product

We agree with the reviewer's suggestion. In the new manuscript, we have added this detail.

(3) Line 80, 'Second, the calculation of radar…'. As discussed in the previous section, it would be more appropriate to refer to it as the "radar correction coefficient" rather than the actual radar penetration. The real radar penetration is dependent on the properties of the snow and ice, not the retracking algorithm.

We agree with the reviewer's suggestion. In the new manuscript, we have modified the description of "radar penetration rate" to "radar correction coefficient".

(4) Line 81, 'Because…, the radar freeboard errors were transferred to the radar penetration rates estimation', this sentence appears somewhat unaware. The empirical method is not the cause of radar freeboard errors or the existence of radar penetration rates.

We agree with the reviewer's suggestion. In the new manuscript, we removed this sentence.

(5) Line 99, '…we used LARM to replace TFMRA…', I don't see why the authors use 'replace' here since there is no consensuses on the algorithm choices until now.

We agree with the reviewer's suggestion. In the new manuscript, we have changed the original expression to emphasize the difference in sea ice thickness introduced between the two radar freeboard products (AWI & Bristol/UIT).

(6) Line 102, 'For the snow depth, we…'. Up until this point, the authors have not highlighted any strengths of the FY3B/MWRI snow depth product. I would suggest that the authors incorporate the benefits of this product in the paragraph discussing snow depth.

In the original manuscript, we described the spatio-temporal availability and accuracy of snow depth data from FY3B/MWRI. In the new manuscript, we have expanded on the snow depth paragraph to provide additional information about FY3B/MWRI.

(7) Line 103, 'Using the three improvements above, we ran four test cases—three individual and one combined…'. Exercise caution when using the term "improvements" when there has been

little discussion of their strengths.

We agree with the reviewer's suggestion. In the new manuscript, we removed the description "improved".

(8) Line 188, 'The difference between AWI CS2 and LARM-derived radar freeboard is mainly due to the different retracking algorithms…', I am pretty sure this is NOT from the Landy et al., (2020), and in fact, what they did is aligning these filtering, corrections and schemes to focus on the effects from retracking algorithm itself. And they continued finding there still exist significant discrepancies from retracking itself. They NEVER said these filtering, corrections and schemes contributed to a relatively small extent. It is definitely sure that classification, waveform filtering, geophysical correction and se level tie-point interpolation exert nonnegligible effects on the final gridded radar freeboard product from each developer.

We agree with the reviewer's suggestion. In the new manuscript, we emphasize that these are two radar freeboard products.

(9) Line 132, 'In this study, the MW99/AMSR2 was used in some optimization cases….', instead of providing a vague explanation, the authors need to clarify where the MW99/AMSR2 dataset was used and the reasons for its inclusion. As of now, it appears that the optimization is limited to the four case studies. Therefore, calling them optimization schemes is questionable, especially considering the authors have not addressed the uncertainty associated with each product. Case studies CANNOT be equated to an optimization scheme.

In section 2.6, titled "Cases of Improvement in Sea Ice Thickness Retrieval," We detailed the use of different snow depth products, where MW99/AMSR2 is the snow depth parameter used in the original AWI CS2. MW99/AMSR2 snow depth is used as the control variable when considering the single effect of radar freeboard and radar penetration rate.

(10) Line 135, I still do not understand why the authors also chose NESOSIM, SnowModelLG, and TOPAZ4, since in Line 102, the authors mentioned the use of FY3B/MWRI. If the authors aim to compare different products to determine the best combination, they should refrain from stating that FY3B/MWRI is used for improvement in the Introduction part.

The purpose of introducing different snow depth products was to complement the Case3, and our goal is to additionally discuss the applicability of these snow depth products to AWI CS2 sea ice thickness retrievals.

(11) Section 2.2, the whole section should have specific description of the spatial and temporal resolution used in this paper, e.g. monthly? Daily? Time span? From which month to month?

It is important to clarify that we describe both the spatial and temporal extent and resolution of these data in Section 2.2.

(12) Line 178, In the Data gridding section, the authors need to explain the data protocol for daily/subdaily datasets (NESOSIM, SnowModel-LG, TOPAZ4, and all observational data) and the monthly dataset (W99/AMSR2, CS-2). They should describe how these datasets are coordinated in this study, such as whether all datasets are averaged into a monthly setting. Additionally, it is important to provide a clear explanation of the method used for spatial interpolation.

In the new manuscript, we add details of the spatio-temporal matching between different data and

a detailed description of the interpolation method (inverse distance weights).

(13) Section 3.1, I have several questions about this section. As I understand it, this section calculates the radar penetration based on all observation radar/snow freeboard and CS-2 LARM radar freeboard, right? In that case, the total freeboard should be calculated from AWI IceBird and IMB ice thickness and snow depth datasets. It is necessary to specify which density is used for these calculations. Furthermore, OIB products have their own protocols for calculating total freeboard. How are these protocols coordinated fairly or placed within the same context? Additionally, since you have already used the results of MYI and FYI penetration factors based on all observations, it is unclear why these datasets are used for further validation. It does not seem fair to use them again for validation, considering they were already used for radar penetration correction.
The original manuscript describes the density parameter in the 2.5 Sea ice thickness retrieval for radar penetration rate calculations (consistent with AWI CS2). Furthermore, in the revised manuscript, the airborne and buoy measurements are no longer used to calculate radar penetration rates. Therefore, potential conflicts between algorithm development and validation datasets would be eliminated.

(14) Line 258, 'The differences in radar penetration rates…'. Once again, it should be noted that the differences in radar penetration can be explained by factors such as frequency, sensor, and period, but not solely by the spatial resolution.
We agree with the reviewer's suggestion. In the new manuscript, we modified this phrase.

(15) Line 259, 'For example, for the OIB, the radar penetration rates may be applicable only in the spring.', so, you did not use the OIB from October to November, right? (That's why the clear information in datasets using in the Data and Method part is very important)
We detailed the spatio-temporal coverage of the OIB data in the original paper (Data and Methods)

(16) Line 267, 'The relationship between FYI and MYI penetration rates supports the previous studies…', It is not clear why you consider all of these relationships to be consistent. Nandan et al. (2017) deduced a depth-dependent saline snow correction factor from observations, and Landy et al. (2022) used 0.9 as a first approximation due to the difficulty of quantifying snow cover changes between May and September. It would be helpful to provide further clarification on how these studies align with your findings.
We have revised the expression of this paragraph.

(17) Fig. 4(a). It is intriguing why the snow depth from FY3B/MWRI is higher in October compared to November. Additionally, it would be beneficial to clarify whether Figure 4 represents Arctic basin-scale mean values. If so, it is puzzling why radar freeboard and thickness are larger in October than in November. Providing possible explanations for these observations would be valuable.
Finding the reasons for explaining these phenomena may be an additional workload beyond the scope of this paper. However, we will try our best to find the causes of these phenomena.

(18) From the Table 2 and Section 3.3, the improvements observed among different cases are only

reflected in the RMSE, which is expected since you corrected or generally reduced the values based on the observations. However, it is frustrating that these four cases differ in at least two products, making it challenging for readers to make direct comparisons.

First, it should be clarified that the single case in the original manuscript changed only one parameter (radar freeboard, radar penetration rate, snow depth) relative to AWI CS2, while the combined case changed all of them. Our baseline is AWI CS2, so this is comparable. Furthermore, we no longer use in situ observations to correct the original radar data in the new manuscript, which makes the validation results referenceable.

(19) Line 313-314, I assume you consider AWI CS2 as your baseline and aim to determine whether the results are better than AWI CS2. If that is the case, you should provide this context from the beginning. However, I have some concerns since the work now uses a completely different algorithm and observed-corrected coefficient for comparison, which may be unfair to AWI CS2. We have modified the original expression.

(20) Line 310-333. Among the in-situ observations, only CryoV ex provides actual independent validation. Upon closer examination of the third column in Figure 6, all cases show high correlation coefficients, and the combination cases reduce the RMSE by over 23% compared to AWI CS2. Therefore, there does not seem to be a significant improvement in the LARM+FY3B/MWRI+RP choice compared to the other cases. The differences lie in the slopes, but it is unclear whether you placed the retrieved data on the x-axis and the in-situ/real data on the y-axis. Mathematically, the x-axis in linear fitting should represent the true/validation data, or else there might be considerable uncertainty in data validation. Therefore, if you were to switch the axes, the slope would likely be different. Additionally, there is a concern that the LARM+FY3B/MWRI+RP combination might result in significant underestimation of sea ice thickness.

Based on the new assessment results of sea ice thickness compared with OIB L4 (Fig. 3) and CryoVEX-EM (Fig. 4), the correlation R value can be kept with the similar results with the original AWI CS2, even in some cases, the R values are higher. We aim to provide some feasible schemes to the optimizations of sea ice thickness derived from AWI CS2. The validation results also did not show that the LARM+FY3B/MWRI+RP combination might result in significant underestimation of sea ice thickness. In addition, based on the principle of regression analysis, we believe that the data used for the X and Y axes (validation and CS2 data) do not affect the final validation results.

[Figure]

Fig 3. Validation of sea ice thickness improvement with the OIB L4. The correlation coefficient (R), root mean square error (RMSE), and the number of samples (N) are shown in each subfigure. The solid black line indicates the best fitting line, and the solid red line indicates the scatter fitting line (the fitting equation is also shown in each subfigure).

[Figure]

Fig 4. Validation of sea ice thickness improvement with the CryoVEX-EM. The correlation coefficient (R), root mean square error (RMSE), and the number of samples (N) are shown in each subfigure. The solid black line indicates the best fitting line, and the solid red line indicates the scatter fitting line (the fitting equation is also shown in each subfigure).

(21) Section 4. It is unclear what the main takeaway is from the entire Section 4, where numerous pictures and discussions focus on the differences between each combination, ranging from spatial patterns to spatial-temporal trends. Since the previous parts have already discussed the improvement in the optimization case, it seems unnecessary to include all combinations here and analyze their differences. This approach might cause readers to lose focus and miss the main points. Additionally, in the abstract, Section 4 is summarized in just one sentence stating that MYI ice thickness is decreasing, which is already quite obvious since lowering the radar correction would naturally reduce the ice thickness. To simplify the paper, it might be better to move Figures 9 and 10 to the Supplementary section.

We agree with the reviewer's suggestion. In the new manuscript, we have simplified the description of the spatio-temporal distribution of the different cases.

(22) Line 455-459, by combining Equations (8) and (9), it is evident that there is a linear relationship between density and radar penetration, with ice density having a larger coefficient than snow density. It would be beneficial to see more uncertainty quantification, such as considering the combined effects of IMB and LARM, and the uncertainties associated with radar penetration derived from observed ice thickness, snow depth, ice density, snow density, and radar freeboard.

We added the sensitivity of the radar penetration rate calculation.

(23) Line 474-477. The temporal sampling is a significant concern in the paper. As mentioned, only IMB data was used from October to February, which raises questions about representativeness and could compromise the results. It would be helpful to provide further explanation on this issue.

In the original paper, we meant to clarify that IMB can be used to calculate radar penetration rate except for March and April (when airborne data are typically derived). We have improved the readability of paragraphs in the revised version.

(24) Figure 15, I am very curious how the radar penetration rates vary from year to year. Including such information in the figure would be valuable.

As mentioned above, we have changed the datasets to calculate the radar penetration rates.

(25) Section 5.2. Like I suggested before, it is important to combine the sensitivities from all parameters. However, it is unclear whether this section focuses on the sensitivity of radar penetration or sea ice thickness. If it is about radar sensitivity, then it is unnecessary to bring up other snow products and their effects, as you have already compared them earlier. It would be better to concentrate on the uncertainties of the FY3B/MWRI snow product in relation to radar penetration. If you also want to discuss the sensitivity study of ice thickness, you should systematically address the uncertainties associated with LARM radar freeboard, FY3B/MWRI snow product, density choice, and derived radar penetration. Additionally, in Figure 16, you introduce another validation on sea ice thickness, which is confusing. It is unclear whether this figure is part of the sensitivity analysis or a validation study.

In Section 5.2 we focus on the applicability of these snow depth products to the AWI CS2 sea ice thickness retrieval. This subsection is an additional analysis of Case3, which also quantifies the effect of snow depth on the sea ice thickness retrieval.

(26) Section 5.3, once again, it is crucial to clearly distinguish between sensitivity studies and validation studies. When discussing the uncertainty of density on sea ice thickness results, it is important to recognize that this pertains to the density choice and its impact on ice thickness. You have already validated the results above and concluded that LARM+FY3B/MWRI+RP is the optimization case. Therefore, please utilize the validated results from earlier and avoid reintroducing these combinations here. Otherwise, it will confuse readers and undermine the confidence and trustworthiness of the previous results. Moreover, it is not appropriate to select densities or refer to them as an "updated density scheme" solely based on having lower RMSE than others after several rounds of validation. We want the paper to avoid cherry-picking results.

Agreeing with the reviewers, in the new manuscript we only analyzed the effect of sea ice density on LARM+FY3B/MWRI+RP. It should be noted that the sea ice density parameters in the original version were derived from the AWI IceBird, which was potentially compared to the A10 to analyze the applicability to the AWI CS2 (Alexandrov et al. 2010, Jutila et al. 2022).

**Reference**

Alexandrov, V., Sandven, S., Wahlin, J., and Johannessen, O.: The relation between sea ice thickness and freeboard in the Arctic, The Cryosphere, 4, 373-380, 2010.

Jutila, A., Hendricks, S., Ricker, R., von Albedyll, L., Krumpen, T., and Haas, C.: Retrieval and parameterisation of sea-ice bulk density from airborne multi-sensor measurements, The Cryosphere, 16, 259–275, 2022.